# Training LLMs over Neurally Compressed Text

**Brian Lester**[a]    **Jaehoon Lee**[b*]    **Alex Alemi**[a]    **Jeffrey Pennington**[a]
**Adam Roberts**[a]    **Jascha Sohl-Dickstein**[b*]    **Noah Constant**[a]
[a] *Google DeepMind*    [b] *Anthropic*
*{brianlester, nconstant}@google.com*

Reviewed on OpenReview: *https://openreview.net/forum?id=pRvhMSV48t*

## Abstract

In this paper, we explore the idea of training large language models (LLMs) over highly compressed text. While standard subword tokenizers compress text by a small factor, neural text compressors can achieve much higher rates of compression. If it were possible to train LLMs directly over neurally compressed text, this would confer advantages in training and serving efficiency, as well as easier handling of long text spans. The main obstacle to this goal is that strong compression tends to produce opaque outputs that are not well-suited for learning. In particular, we find that text naïvely compressed via Arithmetic Coding is not readily learnable by LLMs. To overcome this, we propose Equal-Info Windows, a novel compression technique whereby text is segmented into blocks that each compress to the same bit length. Using this method, we demonstrate effective learning over neurally compressed text that improves with scale, and outperforms byte-level baselines by a wide margin on perplexity and inference speed benchmarks. While our method delivers worse perplexity than subword tokenizers for models trained with the same parameter count, it has the benefit of shorter sequence lengths. Shorter sequence lengths require fewer autoregressive generation steps, often reducing latency. Finally, we provide extensive analysis of the properties that contribute to learnability, and offer concrete suggestions for how to further improve the performance of high-compression tokenizers.

## 1 Introduction

Today's large language models (LLMs) are almost exclusively trained over subword tokens. The tokenizers used to produce these tokens—often BPE (Gage, 1994; Sennrich et al., 2016) or Unigram (Kudo, 2018), as implemented by the SentencePiece library (Kudo & Richardson, 2018)—are compressors that typically achieve ~4× compression over natural language text (Xue et al., 2022).[1] While these tokenizers "hide" the character-level makeup of each token from the LLM (Xue et al., 2022; Liu et al., 2023), this downside is widely seen as outweighed by the significant benefits of compression. Compared to raw byte-level models, an LLM trained over subword tokens sees ~4× more text per token, allowing it to model longer-distance dependencies, ingest more pretraining data, and predict more text at inference time, all without increasing compute.[2]

Given these advantages, it raises the question, could we compress text further to achieve even greater gains? It is well known that autoregressive language models can be turned into lossless text compressors, and recent

---

[*]Work done while at Google DeepMind.

[1]We refer here to "token-level" compression rate, i.e., the length reduction between a raw UTF-8 byte sequence and the corresponding sequence of subword tokens. If instead we measure the number of bits required to encode the two sequences, subword compression typically delivers ~2× or less compression, depending on vocabulary size, which typically ranges from 32k to 256k. See Section 3.4 for discussion.

[2]The increased cost of the input embedding and final softmax layers due to increased vocabulary size is negligible for all but the smallest models.

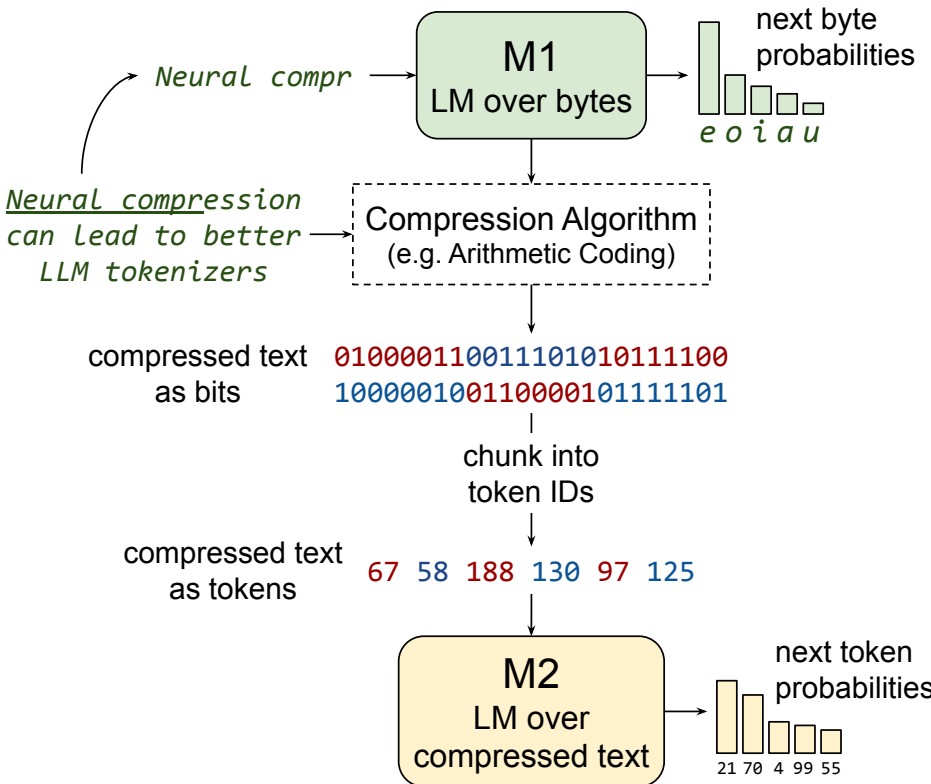

Figure 1: An overview of our approach for training an LLM (M2) over neurally compressed text. First, M1 is trained as a standard byte-level language model—given a leftward context, M1 assigns a probability to each possible following byte. Next, corpus text is compressed into a bitstream using M1 as a compressor. Specifically, the probabilities that M1 assigns at each text position are fed into a compression algorithm like Arithmetic Coding that supports using dynamic symbol probabilities. Finally, this bitstream is chunked into tokens (e.g., 8-bit chunks), and M2 is trained as a language model over compressed text.

work has shown that LLMs can easily achieve 12× compression over English text (Delétang et al., 2024).[3] Can we simply train an LLM over this neurally compressed text?

In this paper we explore various options for doing so, focusing primarily on the idea of using Arithmetic Coding (AC) (Witten et al., 1987), which is known to reach the near-optimal compression rate for a particular model that assigns probabilities to text continuations. Figure 1 presents our high-level approach. First, a small language model "M1" is trained over raw byte sequences. Next, this frozen model is used to compress pretraining corpus text by applying a standard compression algorithm like AC. The resulting compressed bitstream is then chunked into tokens, which are used to train "M2", a language model that directly reads and writes neural-compressed text.

Given a perfect probabilistic model of the raw byte sequence, the compression step would output a fully-compressed bitstream that would be indistinguishable from random noise, and hence unlearnable by M2. In reality, M1 can never be perfect (Zvonkin & Levin, 2007), so the M1-compressed output will still contain learnable patterns. We explore whether using compression powered by a relatively small M1 is able to "remove" the simple structure that M1 understands from the input—e.g., patterns of spelling, word frequency, and basic grammar—while retaining any higher-level structure that M1 fails to model—e.g., patterns requiring "deeper" reasoning and long range coherence. A larger M2 would then learn to model this higher-level structure,

---

[3]Specifically, the authors show that Chincilla 70B (Hoffmann et al., 2022) can compress 2048-byte subspans of enwik9 at a 12× bit-level compression rate.

without needing to relearn the low-level structure removed by M1.[4] In theory, this process could be repeated by training an even-larger M3 model on text compressed by M2, and so on.

In practice, we find that text compressed via Arithmetic Coding is not readily learnable by a standard transformer-based LLM, with resulting models predicting tokens at chance. Interestingly, this result holds even when M1 is reduced to a context-free unigram model, suggesting that the challenge of modeling AC-compressed text stems from the difficulty of learning the AC compression and decompression process itself. We verify this hypothesis by showing that even the sub-tasks of AC-compressing and AC-decompressing text are not learned well beyond a few initial tokens.

To aid learnability, we propose compression via Equal-Info Windows, a simple technique that breaks text into contiguous windows and compresses them via Arithmetic Coding independently. Rather than splitting text into windows of equal text length, we track the number of bits output by the compressor, and close each window just before it exceeds a set information threshold (e.g., 32 bits of information). This has the advantage that when chunking the subsequent bitstream into M2 tokens, there is a stable mapping from N tokens to one window (e.g., four 8-bit tokens ⇒ one 32-bit window). At each window boundary, we reset both AC algorithm and the M1 model context. This ensures that each window may be mapped back onto raw text without any additional information.

Through ablations on window size and M2 vocabulary size, we find that Equal-Info Windows make learning of AC-compressed text possible across a range of settings. However, we also observe that learning progresses gradually, starting with tokens at the left edge of each window, and for longer windows, the model learns little about the tokens near the right edge. Our best-performing setting uses short 16-bit windows that each correspond to a single 16-bit M2 token. Despite resetting the compression algorithm every 16 bits, we still achieve ~5.3× token-level compression overall, which exceeds standard subword tokenizers. Remarkably, our best M2 models outperform byte-level baselines on perplexity benchmarks (bits/byte) for fixed computation budget (FLOPs/byte). This shows that learning over neural-compressed text can be effective.

At the same time, our best M2 models underperform subword baselines. We suspect this is due at least in part to the relatively unstable mappings our neural tokenizers induce between words and tokens. By contrast, standard subword tokenizers induce essentially stable word-to-token mappings, which likely makes the token sequences they output well-suited for LLM training. We illustrate this contrast through qualitative examples. Whether a neural tokenizer can reach a high level of compression while maintaining high learnability for LLM training is an interesting question for future research.

Our main contributions are as follows: (1) Outline advantages and challenges of training over neurally compressed text. (2) Compare LLMs trained over different tokenizers along two axes: bits/byte and FLOPs/byte. (3) Show that standard LLMs can't learn to model vanilla AC-compressed text. (4) Show that GZip-compressed text is learnable by standard LLMs, but not competitive. (5) Propose compression via Equal-Info Windows, and show that it enables learning over neurally compressed text.

## 2 Motivation and Background

### 2.1 Advantages of Training over Neurally Compressed Text

Training LLMs over compressed text is appealing for many reasons. We discuss three advantages in detail below.

**Efficiency**  The most straightforward advantage is efficiency. By compressing the same text into a shorter token sequence, the model can process more text for the same computational cost. In particular, a model trained over $C\times$ compressed text will see $C\times$ more text during training compared to a model trained over raw text, given an equal compute budget. Increasing the amount of data seen in pretraining is often an effective means of improving performance (Kaplan et al., 2020; Hoffmann et al., 2022). Processing text more efficiently also confers benefits at inference time, reducing the serving cost for handling a request of a given prompt and continuation length. In addition to reducing the raw compute needed for inference, compression can also

---

[4]Intuitively, training M2 could be seen as analogous to fitting the residuals of M1 (Friedman, 2001).

improve inference latency, since generating better-compressed output requires fewer sequential autoregressive steps.

**Longer Context** A second advantage is that working with compressed text allows modeling longer contextual dependencies. In vanilla transformer-based models, computation for the self-attention layer scales quadratically with the sequence length, $O(n^2 d)$. This has limited the sequence lengths used by such models in practical settings to ~10k tokens.[5] If, via compression, each token represents (on average) $C$ bytes of raw text, then the resulting LLM can model dependencies across $C\times$ longer distances compared to a raw text model operating over the same token sequence length. While the benefits of modeling longer context (beyond ~1,000 bytes) are modest when viewed merely as perplexity gains (Press et al., 2022), the ability to condition on long context is critical for many applications, such as retrieving content from a document, or answering a coding question provided documentation.

**Distribution of Compute** A third potential advantage of training over compressed text is that information will be spread more uniformly across the sequence. By the nature of compression, a text span that is relatively predictable (e.g., a boilerplate notice) will be more compressible than a span with high perplexity (e.g., a unique product serial number). When an LLM is trained over well-compressed text, each token will represent roughly an equal amount of information. Since the LLM allocates equal compute to each token, this amounts to allocating more compute for "harder" text spans. This adaptivity is similar in spirit to "Adaptive Computation Time" (ACT) (Graves, 2017), which learns to allocate additional compute at some sequence positions in an end-to-end manner, but with the advantage that in our case the computation remains "dense"—identical operations are applied at each position.[6]

## 2.2 Challenges of Training over Compressed Text

**Learnability** It is not at all obvious what types of compression are "transparent" enough to be learnable through a standard LLM training process. Strong compression can be seen as removing as much redundant or predictable information from a sequence as possible. Consequently, the bitstream output by a good compressor is inherently hard to distinguish from random noise. In this work, we explore the setting where M2—the model trained over compressed text—has a larger capacity than M1, the model used for compression. In principle, this setup should allow M2 to extract additional information from the signal even after M1 has compressed it. However, for strong enough M1 compression, the resulting bitstream may be too noisy to detect any signal.

As a prerequisite for M2 to effectively predict continuations of compressed text, we anticipate that it is necessary for M2 to have the ability to decompress bits → text and compress text → bits. These sub-tasks are challenging in their own right. First, M2 needs to accurately "simulate" M1 in order to know the probabilities it assigns to the text, which determine the output of compression.[7] Training models to mimic other models can be difficult (Lester et al., 2022), and even in settings where models do learn to copy the behavior of another network (Hinton et al., 2015), this is often only when looking at which symbol was assigned the highest probability—the actual probabilities assigned often differ (Stanton et al., 2021). Second, M2 needs to learn the compression procedure itself. In our case, this means tracking the Arithmetic Coding algorithm, which requires maintaining high-precision numerical state across long contexts. We investigate these sub-tasks in detail in Section 5.2.

A further learnability challenge is the high level of context sensitivity needed to interpret a bitstream of compressed text. When chunked into tokens, a particular bit subsequence (e.g., `10111001`) can map onto the

---

[5]Exploring sub-quadratic attention mechanisms is an area of active research (Ainslie et al., 2020; Wang et al., 2020; Kitaev et al., 2020; Zaheer et al., 2020; Beltagy et al., 2020; Child et al., 2019, *et alia*). However, regardless of the cost of attention, compressing the input increases the effective context "for free".

[6]It should be noted that ACT learns to allocate more compute where it is *useful*, as opposed to merely where the predictions are hard. For example, ACT learns to not waste compute on inherently unpredictable text spans. We expect that as a heuristic, allocating more compute to higher-perplexity text spans is valuable, but leave this to future work to verify.

[7]For Arithmetic Coding, not only would M2 need to know the probabilities M1 assigns to the observed text, but it would also need to know the probabilities assigned to many *unobserved* symbols. This is because Arithmetic Coding operates over *cumulative* probabilities, i.e., the probability that the next symbol is `e` or any alphabetically preceding symbol.

same token despite having no stable "meaning" across occurrences. We show examples in Section 6.1, where a token maps to many different underlying text forms, necessitating strong contextual understanding. While LLMs are robust to some level of polysemy, as highlighted by the success of Hash Embeddings (Tito Svenstrup et al., 2017) where multiple unrelated words share a single token representation, we suspect this has its limits.

**Numerical Stability**   An additional technical challenge is that compression methods can be sensitive to the precise model probabilities used. To achieve lossless compression in our setup, it is critical that the M1 probabilities match during compression and decompression. This can be hard to guarantee in practice, as there are many sources of numerical noise in LLM inference, especially when running on parallel hardware. An expanded discussion of numerical stability issues can be found in Section 3.7.

**Multi-Model Inference**   Finally, a specific challenge of training over neurally compressed text is that multiple models need to be stored and run side-by-side in order to perform inference. We assume that if M1 is relatively small, this additional overhead is not a significant drawback compared to a standard tokenizer, which is also a separate model that is needed to tokenize text input and detokenize LLM outputs. In evaluating our approach, we include M1 compute in our calculations of total inference cost (FLOPs/byte).

## 2.3   Compression

In this work, we focus on lossless compression, which aims to encode a sequence of input symbols, $x_{0:N} = \{x_0, x_1, \ldots, x_N\} \in X^{|V|}$, into a bitstream while minimizing the expected length of the bitstream. Compression methods are often factored into a "modeling" component and a "coding" component (Mahoney, 2013). The input sequence can be viewed as a sample from a true distribution $p$, $x_{0:N} \sim p$, with a standard autoregressive decomposition, $p(x_{0:N}) = \prod_{i=1}^{N} p(x_i|x_0, \ldots, x_{i-1})$. The "modeling" component aims to approximate $p$ with $\hat{p}$. While some compression algorithms assume static probabilities for each symbol, stronger algorithms are "adaptive", meaning that symbol probabilities may change based on context. In this work, we use context-aware transformer-based language models to represent $\hat{p}$.

The "coding" component of a compression algorithm converts the input sequence to a bitstream of length $\ell(x_{0:N})$. To maximize compression, we want a coding algorithm that minimizes the expected number of bits in the bitstream, $L := \mathbb{E}_{x_{0:N} \sim p}[\ell(x_{0:N})]$. This is done by assigning shorter bit sequences to common symbols and longer sequences to less common ones.[8]   The expected length is lower bounded by $L \geq H(p)$ where $H(p) := \mathbb{E}_{x_{0:N} \sim p}[-\log_2 p(x)]$ (Shannon, 1948). This means that, given a near-optimal coding algorithm, the achievable level of compression derives from how well the model $\hat{p}$ approximates $p$.

## 2.4   Arithmetic Coding

Arithmetic Coding (Rissanen, 1976; Pasco, 1977) uses a model $\hat{p}$ to compresses a sequence $x_{0:N}$ to a bitstream, which is the binary expansion of a float $f \in [0, 1)$. The float $f$ is found by assigning successively smaller sub-intervals to each symbol $x_i \in x_{0:N}$, with the final interval enclosing $f$. An interval is made of an upper and lower bound, $I_i = [l_i, u_i)$ and its size is given by $u_i - l_i$. Starting with $I_0 = [0, 1)$, at each step of encoding, the interval for the symbol $x_i$ is created by partitioning the interval $I_{i-1}$ based on the cumulative distribution of $\hat{p}$ given the previous context, $\hat{p}_{cdf}(x_i|x_{<i})$. The size of this interval is given by $\text{size}(I_{i-1}) * \hat{p}(x_i|x_{<i})$. Thus:

$$I_i(x_i) := \big[l_{i-1} + \text{size}(I_{i-1}) * \hat{p}_{cdf}(w|x_{<i}), l_{i-1} + \text{size}(I_{i-1}) * \hat{p}_{cdf}(x_i|x_{<i})\big),$$

where $w \in X$ is the symbol before $x_i$ in a strict ordering of $X$, i.e., $w$ is the previous token in the vocabulary. Finally, the bitstream of minimal length that represents the binary expansion of a number inside the final interval $f \in I_N(x_{0:N})$ is used as the compressed representation.

---

[8]This process can result in extremely uncommon sequences becoming *longer* under compression, as no algorithm can compress all possible input strings (Mahoney, 2013). In practice, natural language inputs are highly compressible and these edge cases are inputs that one would not recognize as natural language.

Equivalently, the binary expansion can be seen as maintaining a bitstream prefix $b$ and creating successive intervals $B_j(b, x \in \{0, 1\}) := [bl_j, bu_j)$ by partitioning the current interval in half. If the first interval is chosen, a 0 bit is appended to the bitstream prefix $b$, while choosing the second interval appends a 1.

$$B_j(b, 0) := \big[bl_{j-1}, bl_{j-1} + size(B_{j-1}) * 0.5\big)$$
$$B_j(b, 1) := \big[bl_{j-1} + size(B_{j-1}) * 0.5, bu_{j-1}\big)$$

Once the final interval $I_N$ is computed, smaller and smaller bit intervals are created until reaching a bit interval $B_T(b)$ that is fully enclosed by $I_N$. At this point, the corresponding bitstream $b$ is the final compressed representation.

The coding component of Arithmetic Coding is nearly optimal: the output bitstream will have a length of $-\lceil \log \hat{p}(x_{0:N}) \rceil + 1$ bits when using infinite precision. In the finite precision setting using $\beta$ bits, an extra $O(N2^{-\beta})$ bits are added (Howard & Vitter, 1992). See Witten et al. (1987) for an example implementation. In our experiments, we use precision $\beta = 14$. The practical effect of using a finite precision implementation of Arithmetic Coding is that the model's cumulative distribution gets quantized to integers using $\beta$ bits. This results in a minimum probability of $2^{-\beta}$ being assigned to all tokens.

## 2.5 Related Work

Recent work has looked at *using* large language models for compression, but has not to our knowledge attempted to *train* subsequent models over the resulting compressed output. Works like Delétang et al. (2024) use a transformer language model as the modeling component of Arithmetic Coding, but they do not train over compressed output nor do they make modifications to the compression algorithm to facilitate learnability by downstream models. Additionally, they focus on the setting of compressing fixed-size sequences of bytes. By contrast, our models operate over input sequences of fixed *token* length. This allows for models with higher compression rates to leverage longer contexts, as more bytes are included in the input.

Valmeekam et al. (2023) proposes changes to Arithmetic Coding to make it more amenable to use with LLMs—namely, they rank sort the logits from the model before creating text intervals, $I_i(x_{0:N})$. This could help alleviate issues stemming from errors in M2's simulation of M1. However, they do not train models on top of their compressed output.

Some approaches to "token-free" (i.e., purely character- or byte-level) language modeling down-sample the input sequence via convolutions (Clark et al., 2022; Tay et al., 2022), which could be seen as a form of end-to-end neural tokenization. However one important distinction is that the resulting tokenization is "soft"—outputting high-dimensional vectors and not implying a discrete segmentation—in contrast to our tokenization that outputs discrete tokens.

Methods for learning *discrete* tokenization end-to-end have also been proposed (Chung et al., 2017; Godey et al., 2022). In the case of MANTa (Godey et al., 2022), the learned segmentation appears to be fairly semantic (i.e., respecting word and morpheme boundaries), which could be an advantage over our approach. However, they lack our bias towards encoding an equal amount of information per token.

In modeling audio, it is common practice to use learned tokenizers that compress the raw input signal to discrete tokens from a fixed-size codebook (van den Oord et al., 2017; Baevski et al., 2020; Chung et al., 2021; Borsos et al., 2023). However, this compression is lossy, whereas we focus on lossless compression.

Other recent work focuses on using the "modeling" component from well-known compressors to do other tasks. Jiang et al. (2022) uses the model from GZip to perform text classification. Vilnis et al. (2023) uses the Arithmetic Decoding algorithm with an LLM as the model to do diverse parallel sampling from that LLM. One could imagine that the "model" of our compressors (M1) is a teacher for M2, but unlike these other applications, the M1 values are not used outside of compression.

Külekci (2011) also explores learning over compressed text, but with several key differences. First, they use n-gram language models (Shannon, 1948) while we use LLMs. Second, their model is conditioned on

compressed bitstreams but produces a distribution over the raw, uncompressed, bytes while our M2 models predict directly in the compressed space. Additionally, they only consider static Huffman coding (Huffman, 1952) as the algorithm to compress model inputs. While this avoids the context sensitivity issues we outline in Section 2.2, it results in a far worse compression rate compared to the adaptive compression methods we use. One important distinction is that their equal-information windows are overlapping, and used as a sliding window to provide context to their n-gram language model. By contrast our equal-information windows are non-overlapping, and used to segment text into a series of equal-length bitstrings that can be interpreted independently by M2, and whose boundaries are easily identifiable, as they map to a fixed number of M2 tokens.

Several previous works explore how a tokenizer's compression rate correlates with downstream model performance. Goldman et al. (2024) finds that tokenizers that compress better perform better, which generally aligns with our findings, particularly in the large vocabulary setting, see Fig. 6. However, we find that using the *strongest* compressors is detrimental to learnability, as seen by the AC line in Fig. 3.

By contrast, Schmidt et al. (2024) and Dagan et al. (2024) observe a weak trend in the *opposite* direction— better downstream performance from tokenizers that compress *less*. This discrepancy may be due to the fact that Goldman et al. (2024) artificially weakens compression by undertraining the tokenizer, whereas Dagan et al. (2024) compares existing popular tokenizers, and Schmidt et al. (2024) compares different subword algorithms.

The divergences between these results and ours likely stem from major differences in tokenization strategy. These three works are restricted to subword compressors, whereas we explore stronger compressors built on LLMs and Arithmetic Coding. Among other differences, subword tokenizers output tokens in a near-Zipfian distribution (Zipf, 1935), whereas our compressors output a near-uniform distribution, see Table 3. The qualitative differences between these classes of tokenizer are explored more in Section 6.1.

Rajaraman et al. (2024) and Makkuva et al. (2024) find that under some data generation paradigms, transformers fail to model the contextual generation process and instead learn the underlying stationary distribution of the process. This is similar to the failure case we observe where M2 models only output a uniform distribution after training on compressed text. However, their synthetic data setting is quite different from our compressed data setting.

## 3 Methods

For each experiment, we compress long contiguous sequences of training data using different methods. For several, we use M1—a byte-level language model—as $\hat{p}$ in the compression algorithm. We then chunk the compressed output into tokens and train M2 models over those tokens.

### 3.1 Training Data

All training data used is English web text from C4 (en 3.1.0) (Raffel et al., 2020). After tokenization, each document in C4 has an `<EOS>` token appended to it. We concatenate 128 documents together to generate a long sequence of text. Using UTF-8 byte-level tokenization, the average document length is 2,170 bytes, thus these long sequences have an average length of 277,760 bytes. Despite the document breaks, we consider these long sequences "contiguous" for the training of language models. These sequences are then split into individual examples, which are shuffled using the deterministic dataset functionality from SeqIO (Roberts et al., 2023).

### 3.2 Training M1

The model used for compression is a decoder-only Transformer model (Vaswani et al., 2017). It uses the 3m size seen in Table 4 and a context length of 1,024. We use a batch size of 128, an rsqrt decay learning rate schedule ($1/\sqrt{\text{steps}}$) starting at 1.0 with 10,000 warmup steps, and a z-loss of 0.0001. The model is trained for 2,500,000 steps using the Adafactor (Shazeer & Stern, 2018) optimizer. The feed-forward layers use ReLU activations (Nair & Hinton, 2010; Fukushima, 1975), and we use distinct learnable relative attention

embeddings (Shaw et al., 2018) at each layer. We use a deterministic SeqIO dataset and train using Jax (Bradbury et al., 2018), Flax (Heek et al., 2020), and T5X (Roberts et al., 2023). The final validation performance of the M1 model is 1.457 bits/byte, a standard measure of perplexity, see Section 3.8. M1 and M2 are both trained on the C4 training data, but the final validation data used to evaluate M2 is unseen during M1 training, therefore there is no information leakage. This is similar to how LLM tokenizers are often trained on same dataset that the LLM is subsequently trained on.

### 3.3 Compression Methods

When compressing C4 training data, we use an example length of 10,240 bytes and apply one of the following compression techniques (see Appendix B for more methods we considered). This results in compressed examples that are, on average, much longer than our target sequence length of 512 M2 tokens. Thus, each example fills or nearly fills the model's context window with a compressed sequence made from contiguous raw bytes. We compress 51,200,000 examples using each method, allowing us to train each M2 model for 200,000 steps without repeating data.

**Arithmetic Coding**: In this setting, we use a decoder-only transformer language model to model $\hat{p}$, that is, when creating the interval $I_i(x_{0:N})$, the partitions for each possible character, $\hat{p}(x_i|x_{<i})$, are calculated using the probabilities for the next token output by the transformer.

The compressor model is run over contiguous text sequences of 10,240 bytes. The generated logits are used as the model distribution for Arithmetic Coding. We use the Range Encoding (a finite-precision implementation of Arithmetic Coding) implementation from TensorFlow Compression (Ballé et al., 2024) with a precision of 14. The range encoding implementation uses integers with `precision` $+ 2$ bits. This is commonly used when encoding 16-bit float logits, so we do not expect it to cause numerical issues as our models are trained using bfloat16. While the compressor model is only trained on sequences of length 1,024, it uses relative position embeddings in its attention layers. Thus, it can be applied to longer sequences. Some works observe decreased performance as inputs are scaled to lengths beyond those seen in training (Varis & Bojar, 2021; Press et al., 2022), but we find that compression performance is similar in the two settings. Compressing sequences of length 1,024 yields a compression ratio of 5.46 while compressing sequences of length 10,240 yields a ratio of 5.49. This suggests the performance drop from long sequences has minimal effect on compression, or that the increased contextual information makes up this difference.

We will see that text compressed in this straightforward manner is not readily learnable by M2. Thus, we explore alternative compression methods that modify the "modeling" and "coding" components for better learnability. Table 2 shows how our different approaches affect the compression ratio.

**Static Logits Arithmetic Coding**: One potential difficulty of learning over compressed text is that the "modeling" component of the compression algorithm is hard to learn—that is, the second language model (M2) has trouble learning to simulate the probabilities the compressor model (M1) assigns to bytes.

To weaken the compressor model, we replace the context-sensitive LM model with a static byte unigram model—that is, the model's distribution over bytes is the same for each token in the input, i.e., $\hat{p}(x_i|x_0, \ldots, x_{i-1}) = \hat{p}(x_i)$. This distribution is estimated using the byte unigram statistics from the C4 training data.

**Equal Information Windows**: The difficulty in modeling compressed text could also be because the "coding" component of the compression algorithm is hard to learn. That is, the language model is not able to track the state variables used in Arithmetic Coding.

Our proposed method of weakening the coding component of Arithmetic Coding compression is to reset the AC encoder once it has output a set number of bits, creating windows of fixed size where each window is an independently AC-compressed sequence. This process is illustrated in Fig. 2. Windows will represent a variable amount of text, but as each window is created via compression, we expect roughly the same amount of information per window.

In addition to resetting the AC encoder, we also reset the M1 model's context.[9] This means that each $W$ bits of output can be decoded independently, at the cost of a weaker M1 model due to the lack of context.

---

[9]We investigate not resetting M1's context in Appendix A.2 and find that the resets are important for good performance.

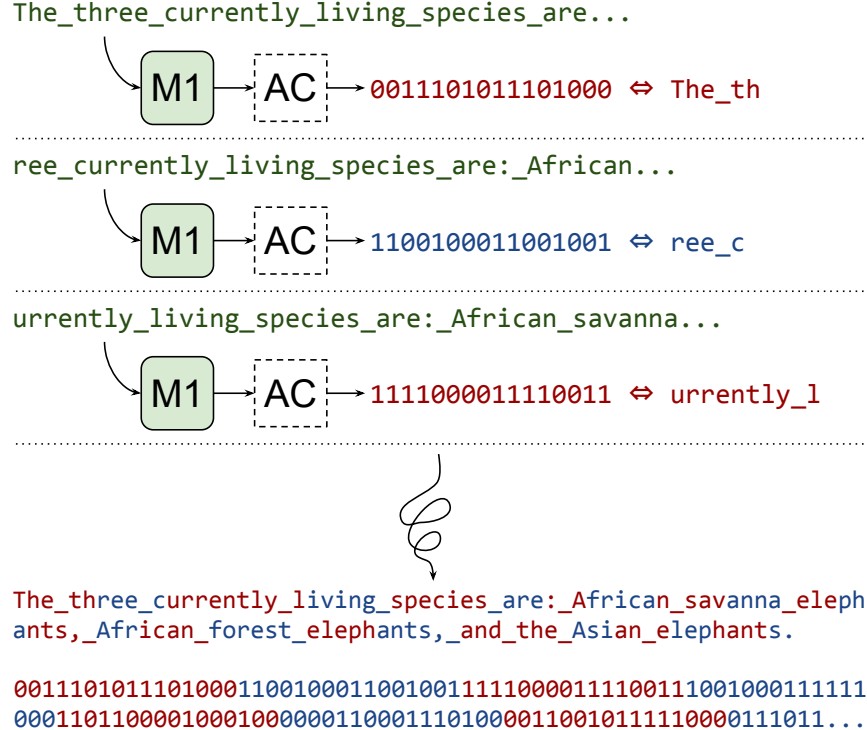

Figure 2: Under "Equal-Info Windows", text is encoded into a series of N-bit windows. To determine each successive window, the remaining text is encoded byte-by-byte via Arithmetic Coding until no more bytes can be added without exceeding the target bit threshold, here 16 bits. Both M1 and the AC algorithm are reset at each step, so no information persists across windows.

As each window is fully self-contained, the model no longer has to learn to track Arithmetic Coding state variables over long distances.

In cases where "spare bits" are available at the end of a window (but not enough to add an additional symbol of text), we pad with zeros. This complicates the decoding algorithm, but the compression scheme can remain lossless. See Appendix D.5 for further discussion, handling the end of the input sequence, and an alternative padding approach that gives similar results.

When compressing an additional character would result in a bitstream that is greater than $W$ bits long, i.e., more than $W$ binary expansions are needed to create an interval that is enclosed by $I_{i+1}(x_{0:i+1})$, the bitstream (padded to $W$ bits as necessary) representing the input up to and including character $i$ is emitted. Then both the AC encoder and M1 model are reset. That is, $I_{i+1}(x_{i+1:N})$ is calculated as if $I_i(x_{0:i}) = [0, 1)$; the bit interval is also reset to $B_j(b = "") := [0, 1)$. Similarly, M1 is only conditioned on inputs that are part of the current window, the inputs after $i$. That is, $\hat{p}(x_j|x_{<j}) \approx \hat{p}(x_j|x_{i...j})$.

We use $b$ to denote the bits per window, and $v$ for the vocabulary size of M2. For example, EqualInfoAC[$(b)its$=16, $(v)ocab$=256] represents AC encoding with 16-bit Equal Info Windows and 8-bit M2 tokens (vocabulary 256).

**GZip**: As a baseline, we also explore training over text compressed using GZip (Deutsch, 1996) as implemented in the Python (Van Rossum & Drake, 2009) `zlib` library using the default compression level. GZip uses the `DEFLATE` algorithm—a combination of Huffman Trees (Huffman, 1952) and LZ77 (Ziv & Lempel, 1977). First LZ77 is used to replace repeated substrings in the text with pointers back to the original substring. Then a Huffman Tree is built for the current—LZ77 compressed—example and used to compress it. Note that this setting is dynamic, as the Huffman tree, and hence the binary codes for each character, are unique

Table 1: "Token" vs. "bit" compression ratios. Larger vocabularies require more bits to store each token, and thus incur a cost in terms of absolute compression. However, when trying to minimize the compute an LLM uses to process a given piece of text, token sequence length is what matters.

| Method | Token Compression Ratio | Bit Compression Ratio |
|---|---|---|
| SentencePiece | 4.28 | 2.28 |
| AC[$v$=256] | 5.49 | 5.49 |
| AC[$v$=65k] | 10.98 | 5.49 |

to the example. These experiments explore a setting where both the modeling and coding components of compression are different from Arithmetic Coding.

### 3.4 Tokenization of Compressed Text

Most compression methods output a bitstream, but training M2 directly over bits would not be ideal. As M1 was trained over UTF-8 bytes, the bit-level output of compression would result in M2 being applied to much longer sequences. Additionally, models are generally trained with vocabulary sizes much larger than two. Thus, we need a method to segment the bitstream into tokens, creating a more standard sequence for training language models.

We convert the bitstream into a token sequence by grouping every $N$ bits into a token—resulting in a vocabulary size of $2^N$. We explore settings of $N \in \{8, 16\}$, resulting in vocabulary sizes of $v$=256 and $v$=65,536. As the tokens are created from the compressed bitstream, we expect the distribution of tokens to be more uniform than the usual Zipfian (Zipf, 1935) distribution of word or subword tokens, allowing us to use larger vocabularies without encountering issues of rare or unattested tokens.

Following Rajaraman et al. (2024), our tokenization scheme can be described as the tuple $\mathcal{T} = (\text{Dict}, \text{DS}, \text{enc}(\cdot), \text{dec}(\cdot))$. Dict is the space of tokens created by segmenting the compressed bitstream— in our case, a vocabulary of 256 or 65k. DS is the entire M1 model. The functions $\text{enc}(\cdot)$ and $\text{dec}(\cdot)$ perform encoding (compression) and decoding (decompression). In our case, these functions vary with (i) the M1 model, (ii) the compression algorithm (AC, EqualInfoAC, etc.), and (iii) the bitstream segmentation strategy.

Throughout this work, we focus on the "token compression ratio" $L_{iT}/L_{oT}$—the ratio between the input and output token sequence lengths. It is important to note that the meaning of "token" can differ between the input and output sequences. Generally, the input sequence is one byte per token, while output tokens represent multiple bytes. This is in contrast to the more standard "bit compression ratio" $L_{ib}/L_{ob}$—the ratio of input bits to output bits. As we aim to reduce the computational overhead of running LLMs by training them on compressed input, we are more concerned with reducing the number of tokens that M2 consumes. This difference is elucidated in Table 1. While SentencePiece results in a sequence length reduction of 4.28×, the larger vocabulary means that 15 bits are required to represent each token. As such, the bit compression ratio is only 2.28, which is much lower than our AC-based compressors. Similarly, creating 16-bit tokens from the output of Arithmetic Coding does not change the bit compression ratio—the total number of bits is unchanged—but it does reduce the number of tokens in the sequence, and thus the number of tokens the LLM must process. We compute compression ratios over the C4 dev set, which is unseen during M1 training.

To highlight the differences between the tokenization methods above, we measure the performance (as bits/byte on a sample of the C4 validation set) of two trivial models for each tokenizer in Table 3. The "uniform" model naïvely assigns equal probability to each token, regardless of context. The "unigram" model also ignores context, but assigns probabilities based on the global token frequencies observed in the training data. With byte-level tokenization, each UTF-8 byte encodes to a single 8-bit token, so the uniform model achieves 8 bits/byte. For more powerful tokenizers, the uniform model is stronger, indicating that the tokenizer itself has some language modeling ability. We observe that our compression-based tokenizers (AC, EqualInfoAC and GZip) output a near-uniform distribution of tokens across their vocabulary. This is reflected in the near-zero gain over "uniform" achieved by modeling unigram statistics.

Table 2: Weakening the "model" or "coding" component of Arithmetic Coding reduces the compression rate. The reduction of M1 to a static unigram distribution results in the worst compression ratio. When using EqualInfoAC, M1 is weaker, as it has less context, and coding is weaker, as padding is often required at the end of windows. The compression ratio improves with larger window sizes.

| Method | Compression Ratio |
|---|---|
| AC[$v$=256] | 5.49 |
| StaticAC[$v$=256] | 1.73 |
| EqualInfoAC[$b$=16, $v$=256] | 2.66 |
| EqualInfoAC[$b$=32, $v$=256] | 3.49 |
| EqualInfoAC[$b$=64, $v$=256] | 4.16 |
| EqualInfoAC[$b$=128, $v$=256] | 4.61 |

Table 3: Bits/byte ($\downarrow$) performance of two trivial models across tokenizers. "Uniform" assigns equal probability to each token. "Unigram" assigns probabilities based on the empirical token frequencies. As the compression-based tokenizers output near-uniform distributions over tokens, there is little gain in modeling unigram statistics. Thus, learning over this data requires modeling longer contexts.

| Method | Uniform bits/byte | Unigram bits/byte | $\Delta$ |
|---|---|---|---|
| Bytes | 8.000 | 4.602 | 3.398 |
| SentencePiece | 3.497 | 2.443 | 1.054 |
| AC[$v$=256] | 1.457 | 1.457 | 0.000 |
| StaticAC[$v$=256] | 4.624 | 4.624 | 0.000 |
| EqualInfoAC[$b$=16, $v$=256] | 3.008 | 2.976 | 0.032 |
| EqualInfoAC[$b$=32, $v$=256] | 2.292 | 2.285 | 0.007 |
| EqualInfoAC[$b$=64, $v$=256] | 1.923 | 1.921 | 0.002 |
| EqualInfoAC[$b$=128, $v$=256] | 1.735 | 1.735 | 0.000 |
| GZip[$v$=256] | 3.587 | 3.586 | 0.001 |

### 3.5 Training M2 on Compressed Data

Each M2 model is trained for 200,000 steps with a batch size of 256 and a sequence length of 512. Thus each model trains on 26.2 billion tokens. Of these, the vast majority (over 98.9%) are non-padding tokens; see Appendix D.2 for details and Table 12 for the exact size of each dataset. As methods with higher compression ratios cover more raw text per token, we also include the total number of bytes in each dataset. Shuffling of training sets is seeded, and dataset state is checkpointed during training, so each training run results in the model seeing each example exactly once.

Models are trained at four sizes, as shown in Table 4, with 25m, 113m, 403m, and 2b parameters, excluding embedding parameters. When the compressed bitstream is chunked into 8-bit tokens, the M2 model has a vocabulary size of 256. With 16-bit tokens the vocabulary increases to 65,536. All M2 models have a sequence length of 512 tokens. Thus, when training on 16-bit tokens, twice as many bytes are seen per example and in training overall, as compared to 8-bit tokens. All other hyperparameters match those used in M1.

### 3.6 Baselines

We compare our M2 models against baseline models trained with two standard tokenization methods, described below. All hyperparameters, including sequence length (512), match those used for our M2 training above.

**Bytes:** These baselines train directly over UTF-8 bytes, using the byte tokenizer from ByT5 (Xue et al., 2022), which simply encodes the input string as UTF-8 using Python `s.encode("utf-8")`. The models see 26.2 billion bytes total (see Table 12).

Table 4: Model sizes used in our experiments, and corresponding hyperparameter settings. Note, model parameter counts exclude embedding table parameters.

| Parameter Count | Embedding Dim | #Heads | #Layers | Head Dim | MLP Dim |
|---|---|---|---|---|---|
| 3m | 256 | 4 | 3 | 64 | 1024 |
| 25m | 512 | 8 | 6 | 64 | 2048 |
| 113m | 768 | 12 | 12 | 64 | 3072 |
| 403m | 1024 | 16 | 24 | 64 | 4096 |
| 2b | 2048 | 32 | 24 | 64 | 8192 |

**SentencePiece:** These baselines train on text tokenized using the T5 (Raffel et al., 2020) vocabulary, which has 32,000 tokens, and was trained using the Unigram algorithm (Kudo, 2018) as implemented by the SentencePiece library (Kudo & Richardson, 2018). Unigram is one of several popular subword tokenization algorithms, with others including BPE (Sennrich et al., 2016) and WordPiece (Schuster & Nakajima, 2012; Wu et al., 2016), and each of these is implemented by multiple libraries, sometimes with minor differences.

We opt to use the SentencePiece Unigram tokenizer as our subword baseline for a few reasons. First, the performance of these competing subword tokenization algorithms is known to be very similar, with several works finding no statistically significant difference between them in many settings (Kudo, 2018; Ali et al., 2024; Schmidt et al., 2024). Second, where differences are reported (marginal or otherwise), Unigram has been found to be preferred, at least for English-only models (Kudo, 2018; Bostrom & Durrett, 2020; Ali et al., 2024; Schmidt et al., 2024). Finally, previous work indicates that the SentencePiece implementation of Unigram outperforms the HuggingFace implementation (Ali et al., 2024; Schmidt et al., 2024).

Our baseline models trained over SentencePiece tokens see 112 billion bytes total (see Table 12).

## 3.7 Numerical Stability

Arithmetic Coding depends on the creation of "intervals" that cover each symbol in the vocabulary based on the quantized cumulative distribution of a model's logits when predicting the next token. As such, a small change in the logits due to numerical noise can result in vastly different output bitstreams. This can make the practical use of neural language models in compression difficult. Common sources of noise include changes in batch size, parallel computation, changes to compute infrastructure (CPU vs. GPU vs. TPU, different TPU topology, etc.), changes to inference (computing the logits for the whole sequence at once vs. computing logits for a single token at a time using KV caches), and changes to the longest sequence length in the batch.

Methods like the rank-sorted algorithm used in LLMZip (Valmeekam et al., 2023) may help alleviate these issues as only the order of tokens needs to match between settings. The development of alternate methods of LLM-based compression should keep numerical stability issues in mind and ideally alleviate these issues in the design of the algorithm. Increasing the level of quantization could also help reduce numerical noise issues, as differences would mostly be lost in quantization, but this would have a negative impact on the compression ratio.

## 3.8 Evaluation

As the tokenization scheme varies across the approaches we consider, models cannot be directly compared on "per-token" metrics such as negative log likelihood loss $\ell$. Rather, following previous work (Dai et al., 2019; Al-Rfou et al., 2019; Choe et al., 2019; Gao et al., 2020, *et alia*), we report perplexity in terms of "bits-per-byte", $[\text{bits/byte}] = (L_{oT}/L_{iT})\ell/\ln(2)$, which scales the model's loss by the token-level compression rate.

We also compare models on how much computation (FLOPs) is required to perform inference over a given length of raw text (bytes). More specifically, we calculate M2's expected FLOPs/byte by scaling FLOPs/token—approximated by $2 \times$ `params` (excluding embedding parameters) following Kaplan et al. (2020)—by the

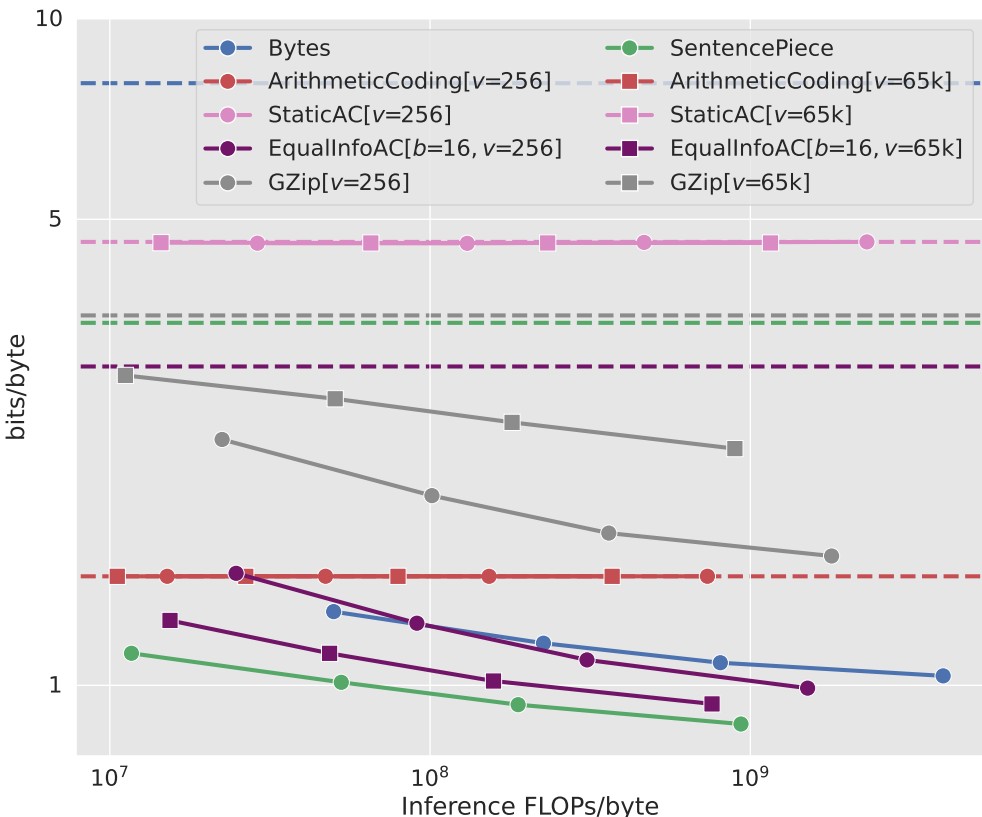

Figure 3: Models trained over compressed text are compared against baseline models in terms of bits/byte (↓) and inference FLOPs/byte (↓). The ArithmeticCoding and StaticAC settings are essentially unlearnable, with models failing to outperform naïve baselines (dashed lines) that assign equal probability to all tokens. EqualInfoAC and GZip outperform naïve baselines and show improvement with scale. EqualInfoAC is the strongest of the compression-based methods, with EqualInfoAC[$b$=16, $v$=65k] outperforming the Bytes baseline at all sizes. While SentencePiece performs the best, the gap between EqualInfoAC and SentencePiece narrows with scale. See Appendix D.9 for the exact values used in this and other graphs.

token-level compression rate (as tokens/byte). For methods using an M1 model during compression, the FLOPs/byte cost of M1 is added.[10] For more details on the evaluation metrics see Appendix D.4.

We evaluate models on a sample of the C4 validation set. During evaluation, the model is run over 20 batches or ~2.6 million tokens. These tokens represent different amounts of text based on the compression method, making it impractical to run evaluation on the same sequence of bytes for all methods. To confirm that our validation samples are large enough to be representative, for each method, we train five 25m parameter models with different seeds. We find the final performance to be extremely stable, with the largest standard deviation in bits/byte being 0.0061. Thus, the variance introduced from sampling the validation set is negligible. See Appendix D.1 for more information about variance.

## 4   Results

**Simple Methods of Training Over Neurally Compressed Text Fail**   As seen in Fig. 3, the most obvious approach—compression using Arithmetic Coding with M1 assigning next-token probabilities—fails to learn anything. Regardless of scale, the model only learns to output a uniform distribution over tokens, the

---

[10]While there is a computational cost to running GZip over the input text, we ignore it as it is insubstantial compared to the cost of running M2 model inference.

performance of which is denoted by the dashed line. As the Arithmetic Coding procedure is near optimal (Mahoney, 2013), the compression ratio is essentially determined by the loss of M1. Thus, even though the M2 model learns nothing useful, when scaled by the compression rate, this setting ends up with the same performance as the M1 model. Similarly, models trained over data compressed with StaticAC—where M1 is replaced with a static unigram model—fail to learn. This result suggests that the difficultly in learning stems from the complexity or brittleness of the Arithmetic Coding process itself, rather than from M2's inability to model M1. Note that the weak "modeling" component of this compression scheme results in a much lower compression rate and thus worse bits/byte performance, despite the model also learning a uniform distribution.

On the surface, M2's inability to learn seems to run counter to Yun et al. (2020), which finds that transformers are universal approximators of sequence-to-sequence functions. However, the fact that transformers can *express* a function does not yet imply the ability to *learn* that function via stochastic gradient descent from random initialization. Additionally, for the purpose of training LLMs, our interest is in training a model that generalizes well to held-out validation data, as opposed to one that overfits its training data. In practice, we find our M2 training procedure *is* sufficient to overfit to small subsets of our AC-compressed training data, when repeated over multiple epochs. However, this does not satisfy our goal of finding a compression scheme suitable for LLM training.

**SentencePiece is a Strong Baseline**   Our SentencePiece baseline outperforms all other methods, including our Bytes baseline, across all model sizes. On the surface, this result seems to run counter to the recent findings of Delétang et al. (2024), where their byte-level models outperformed subword (BPE) models at medium and large scales. The discrepancy is due to prioritizing different metrics. They report the model's bit compression rate on fixed-length (2,048 byte) sequences. While this is one type of "fair" comparison, it disadvantages subword models, as they are trained to model dependencies longer than 2,048 bytes (but never evaluated on this ability), and are allotted fewer inference FLOPs to process the same text, as compared to the byte-level models. Additionally, *bit* compression ratio penalizes subword models for having larger vocabulary sizes. By contrast, our evaluation tests what perplexity models achieve on sequences of the same length they were trained on, and compares models at matching FLOPs/byte cost. This aligns with our end goal, which is to train an LLM that achieves the best perplexity at whatever sequence length it can handle, given a fixed budget for training and inference.

**Equal-Info Windows make AC Learnable**   Fig. 3 shows that EqualInfoAC[$b$=16, $v$=256] outperforms the byte-level baseline at most model sizes, with the gains increasing with scale. In addition to better bits/byte performance, training over compressed data has the advantage of using fewer FLOPs/byte for a given model size—seen in the leftward shift of the EqualInfoAC[$b$=16, $v$=256] curve compared to the Bytes curve—due to shorter sequence lengths.

Using 16-bit tokens (65k vocabulary) increases performance further. EqualInfoAC[$b$=16, $v$=65k] outperforms the Bytes baseline at all model sizes. It underperforms the SentencePiece baseline, but the gap diminishes with scale.

However, EqualInfoAC[$b$=16, $v$=65k] *outperforms* the SentencePiece baseline in terms of tokens/byte. Models using EqualInfoAC[$b$=16, $v$=65k] take fewer autoregressive steps to generate the same text than models using SentencePiece encoding. This has the potential to reduce generation latency, at the cost of reduced compute efficiency. This is a tradeoff that is often worth making in production. For instance, speculative decoding (Leviathan et al., 2023) is a popular approach that performs redundant computation in order to potentially accelerate auto-regressive steps.

It is noteworthy that the EqualInfoAC M2 models learn well despite being trained on data that has nearly uniform unigram statistics, as we saw in Table 3. In the best case, our 2 billion parameter M2 model achieves 0.94 bits/byte. This is a large gain over the naïve uniform (3.01 bits/byte) and empirical unigram (2.98 bits/byte) models from Table 3, and approaches the performance of a parameter-matched SentencePiece model (0.87 bits/byte), despite using 23% fewer FLOPs/byte.

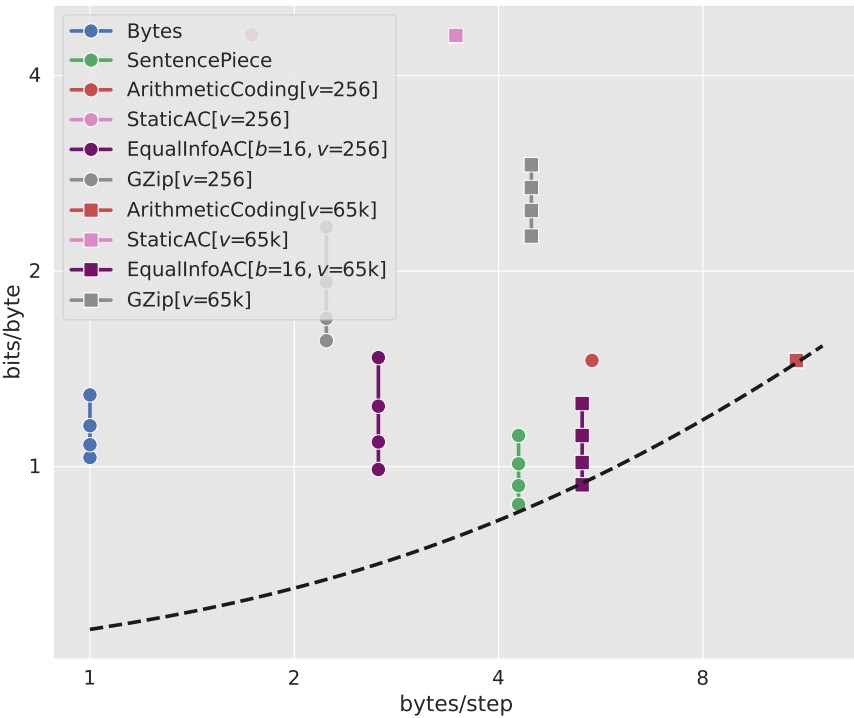

Figure 4: Comparing models in terms of bits/byte (↓) and bytes/step (↑). As decoder steps can be a practical bottleneck for system latency, a model with higher FLOPs/byte or worse bits/byte may be preferred in order to achieve shorter sequence lengths. The dashed line (– ·) is an example Pareto frontier, showing how a practitioner might value the trade-off between bits/byte and bytes/step. Our 2 billion parameter EqualInfoAC[$b$=16, $v$=65k] model is on this frontier.

It is apparent from Fig. 3 that if FLOPs/byte were held constant, SentencePiece would achieve slightly better bits/byte than EqualInfoAC. However there is another axis along which EqualInfoAC may still be preferred. Setting aside inference FLOPs, on average SentencePiece tokenization requires 23% longer sequences to encode the same text when compared to our best EqualInfoAC setting ($b$=16, $v$=256). This means that, regardless of FLOPs used, the SentencePiece models will take more decoder steps at inference time. It is up to the practitioner whether it is "worth it" to trade off some bits/byte performance in order to achieve shorter sequences. In many serving scenarios, decoder steps are a practical bottleneck for determining system latency, as other aspects of model scale, such as width, can be mitigated with enough parallelism. There are cases where one may be willing to incur even more (parallelizable) inference costs to reduce latency, as in "speculative decoding" (Leviathan et al., 2023). To this end, it may be advantageous to use more compute resources to scale up an EqualInfoAC[$b$=16, $v$=65k] model (recovering bits/byte performance) while retaining the reduced latency due to the shorter sequence length. This can be seen visually in Fig. 4.

**GZip is Not Competitive**   Training over GZip-compressed text is relatively ineffective. M2's performance when trained over GZip highlights a counter-intuitive trend. While the GZip M2 models actually learn, it would still be preferable to train over AC-compressed text—even though those models do not learn. This is due to the weak compression offered by GZip. The poor compression rate, coupled with weak learning, means that the GZip M2 models' bits/byte performance lags behind even the 3m parameter M1 model.

**Short Windows are the Best**   We see a similar effect in Fig. 5, which ablates the EqualInfoAC window size. In terms of bits/byte, the shortest 16-bit windows perform the best. However, the next-best setting is the longest 128-bit windows, despite the fact that these M2 models fail to learn almost anything beyond the uniform distribution. This unintuitive trend stems from the fact that longer windows translate to better compression rates (see Table 2). If we remove the effect of compression rate by looking at bits-per-token

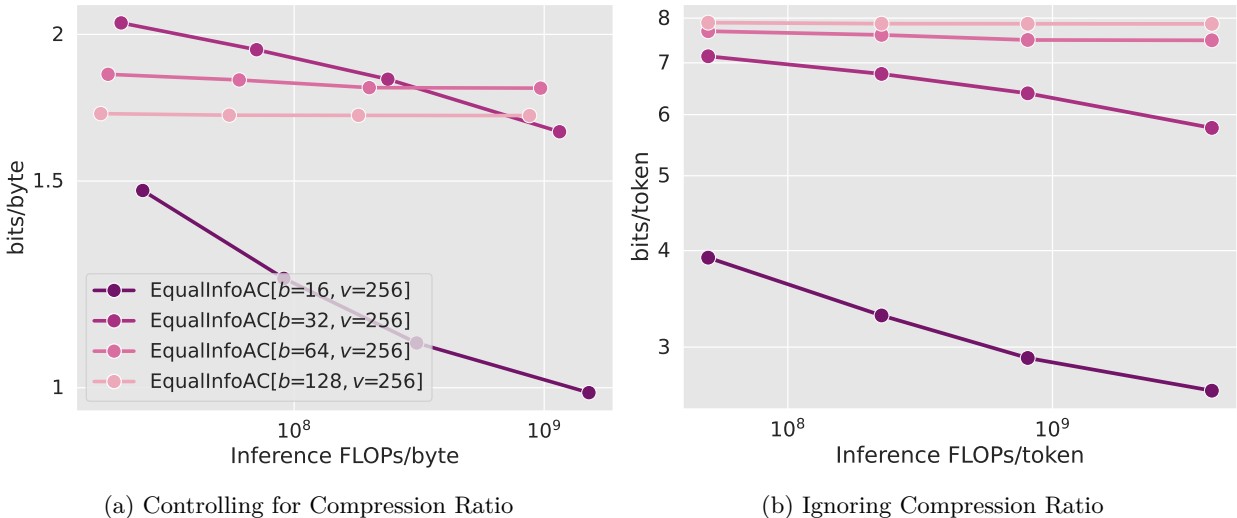

(a) Controlling for Compression Ratio

(b) Ignoring Compression Ratio

Figure 5: Performance of EqualInfoAC across various window sizes, $b \in \{16, 32, 64, 128\}$. When evaluating bits/byte (left) to control for compression ratio, we see an unintuitive trend where for most model sizes $b = 16$ is best but $b = 128$ is second-best. This is due to the higher compression rate achieved by longer Equal Info Windows. When evaluating tokens/byte (right), a monotonic trend emerges, showing that shorter windows are easier to learn.

(Fig. 5b), we see a clearer monotonic trend—increasing window length makes it harder to learn, as we move closer to simply running Arithmetic Coding over the whole sequence. For 64 and 128-bit windows, performance improvements with scale are small, but present; see Table 16 for exact numbers.

**Larger M2 Vocabulary is Helpful** Tokenizing compressed text using a larger 16-bit vocabulary ($v$=65k) results in a 2× higher token compression rate, seen in the leftward shift of each curve in Fig. 6.[11] For Arithmetic Coding methods, larger vocabulary also improves bits/byte, seen as a downward shift in the curves. However, for GZip, we see the opposite trend. Arithmetic Coding and GZip differ the most in their coding component, which suggests that the reason for this difference could lie there. Note that the header and footer present in GZip-compressed data do not explain this difference, see Appendix A.4. For EqualInfoAC[$b$=16], moving from $v$=256 to $v$=65k results in each window corresponding to a single token, which increases the "stability" of the token → text mapping. This could be one reason for the performance gain; see Section 6.1 for more discussion of "stability".

**Emergence with Scale is Unlikely** Given the recent findings of Schaeffer et al. (2023), we anticipate that continuing to scale models beyond 2 billion parameters is unlikely to deliver an "emergent" ability to learn over AC-compressed text, since the bits/byte metric we use is smooth.

**Results Persist Under "Scaling Laws" Paradigm** When scaling models, Hoffmann et al. (2022) recommend that training tokens should be scaled linearly with model size. However, in our experiments above, all models see the same number of tokens, regardless of model size. Consequently, our largest models may be somewhat "undertrained".[12] To test whether following the "scaling laws" recommendation influences our results, we reevaluate our models at earlier checkpoints selected to maintain a constant ratio of training data to model size. We find that all core trends are unchanged in this setting. See Appendix D.3 for details.

---

[11]The same trend holds for larger 64 and 128-bit windows, but the performance increase with scale is so slight that we omit them from the graph. See Table 16 for the exact values.

[12]The undertraining of our 2b models is also visible in their validation loss curves, which still have a significant decreasing slope at 200,000 steps, showing the models have not yet converged.

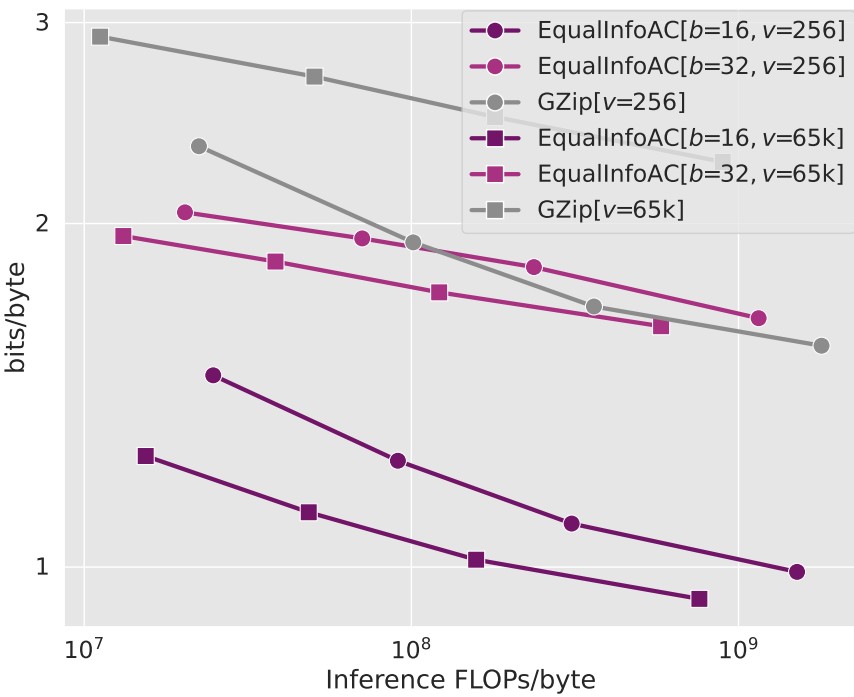

Figure 6: Using a larger vocabulary for Arithmetic Coding derived methods improves both perplexity (lower bits/byte) as well as token compression rate (lower FLOPs/byte). Among settings where the M2 model actually learns, training over GZip-compressed data is the only case where increasing vocabulary size to 65k does not help performance.

## 5 Additional Experiments

At this point, we have established that while the simplest approaches to training over compressed text fail, there are alternate compression schemes that are learnable. In this section, we conduct additional experiments to shed light on which aspects of different compression methods are difficult to learn and what contributes to their learnability.

### 5.1 Bitstream Tokenization is Not the Main Source of Difficulty

The compression algorithms we consider output a bitstream, which we later chunk into tokens of a fixed bit depth (e.g., 8-bit tokens). As such, it is common for the bits representing a single character or UTF-8 byte to be split across multiple tokens. Compounding this issue is that the value of these tokens are contextually determined and may differ depending on the surrounding bytes.

The fact that both 8-bit and 16-bit token chunking strategies work suggests that this is not too much of an issue for the model. To further investigate this, we train two models—one 25m and one 403m—on the raw bitstream output by Arithmetic Compression, i.e., each token is either a 1 or a 0 and the vocabulary has a size of 2. We use the same hyperparameters as in Section 3. Working at the bit level means that the output sequence is now *longer* than the input sequence, which was UTF-8 bytes. As such, this setting is not practical in the real world.

When trained to convergence, the two models have cross entropy losses of 0.693 for the 25m parameter model and 0.6928 for the 403m model—not meaningfully better than the naïve uniform distribution, which yields a loss of 0.693. This failure mode is the same as in Fig. 3, which suggests that AC encoding itself is the main source of difficulty, as opposed to any issue around tokenization or vocabulary size.

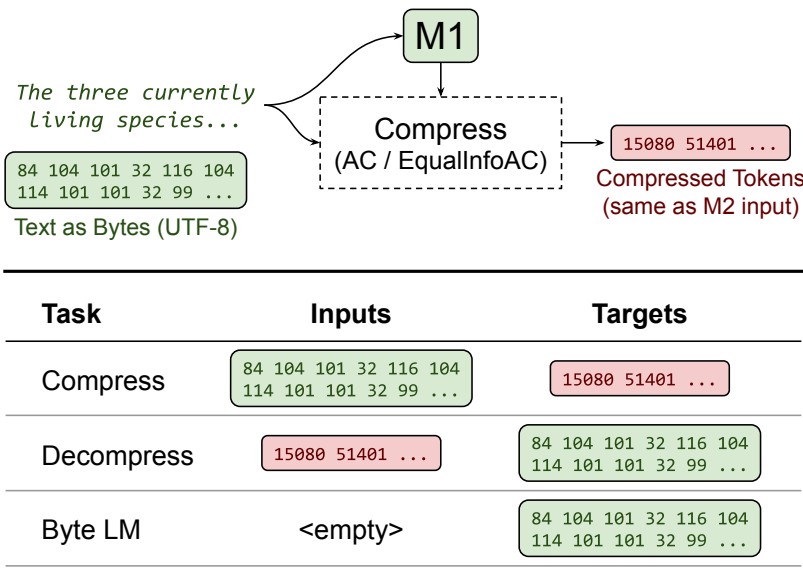

Figure 7: Arithmetic compression and decompression cast as sequence-to-sequence tasks. To account for the strong performance that is possible by modeling only the output bytes in the decompression task, we also train a Byte LM on *just* the targets.

Table 5: Transformers struggle to learn Arithmetic Coding. In the sequence-to-sequence setting, a model that learns AC compression/decompression should have an accuracy of 100. Our models perform much worse. When tasked with decompression in a sequence-to-sequence format, our transformer's improvement over pure language modeling of the targets was not statistically significant ($p = 0.07$). Thus, the model is not able to leverage the compressed input. Similarly, AC compression is only learned to 1.7% accuracy.

| Compression | Task | Accuracy | Cross Entropy |
|---|---|---|---|
| AC[$v$=256] | Decompress | 76.98 | $0.751 \pm 0.005$ |
| | Byte Level LM | 76.86 | $0.755 \pm 0.001$ |
| EqualInfoAC[$b$=16, $v$=256] | Decompress | 95.30 | $0.141 \pm 0.002$ |
| | Byte Level LM | 76.83 | $0.756 \pm 0.001$ |
| AC[$v$=256] | Compress | 1.7 | 2.489 |

## 5.2 Transformers Struggle to Learn Arithmetic Coding

Arithmetic Coding is a sequential algorithm that involves tracking multiple state variables as the input (byte) sequence is consumed. Each token in the output sequence represents multiple transformations of these variables, e.g., 8 transformations when using 8-bit token chunking. Theoretically, only 10 transformer layers are needed to have a computational path through the model layers that can process a sequence of 1,024 tokens as a chain, where each token conditions on the previous one. While most of our transformers have the capacity to model these sequences—only our 25m model has fewer layers—we see in practice that the Arithmetic Coding algorithm is still difficult to learn.

To directly diagnose the ability to track Arithmetic Coding, we format AC compression and decompression as sequence-to-sequence tasks, as shown in Fig. 7. The input provides the model with the true text, so we expect a model that is able to learn Arithmetic Coding should achieve an accuracy of 100. We compress sequences of 1,024 bytes using M1 and Arithmetic Coding.[13] We concatenate the bytes and AC output tokens to create the compression task. For the decompression task, we simply flip the order—AC output tokens first and then bytes. The target token IDs (bytes or tokens) are shifted by the input vocabulary size, ensuring that they

---

[13]We use shorter raw text sequences to keep the final sequence length of inputs + targets manageable.

have distinct values. We use a decoder-only transformer as our model with a causal attention mask, i.e., even during the input sequence, future tokens are hidden from the model. We train models with 113m parameters. Loss, gradients, and evaluation metrics are only computed on the target tokens.

In the decompression task, the target tokens are bytes. By ignoring the inputs and just modeling the outputs, the decompression model can achieve decent performance without actually leveraging the input data. To control for this, we also train a byte-level language model baseline on the same sequence-to-sequence data, excluding the input tokens. If the decompression model is actually learning to decompress Arithmetic Coding, we would expect stronger performance than the byte-level baseline. As we see in Table 5, the baseline model, which does not see the input tokens, has the same performance as the decompression model.[14] Clearly, the models trained for decompression are not actually learning to do decompression. In contrast, we see that models trained on EqualInfoAC[$b$=16, $v$=256] do appear to learn to perform decompression. An M2 model trained to decompression EqualInfoAC[$b$=16, $v$=256] compressed data quickly jumps to 90% accuracy, in 9K updates, and continues to improve to 95% over training. The model is still improving as training ends with no clear signs of overfitting, suggesting the expected 100% accuracy should be achievable with more data.[15] It clearly leverages the input as a byte-level language model trained over the same target tokens only reaches an accuracy of 76.

The model trained for compression actually shows some signs of learning. Training a language model directly on the compressed output results in the model learning a uniform distribution over tokens, see Fig. 3. When the model is able to attend to the input text, we see that the performance in Table 5 is better than the uniform distribution (which would have a cross entropy loss of 5.545). While this method shows some hope for the learnability of Arithmetic Coding, the need to include the input sequence negates the main advantage of compression, i.e., applying the model to a shorter sequence. Additionally, the compressor's performance is far from the 100 it should be able to achieve.

We also find training on these sequence-to-sequence datasets to be less stable than training on the language modeling datasets. In our experiments, large performance swings and divergence were relatively common.

### 5.3 Larger Vocabulary Helps Beyond Increasing the Compression Ratio

Our best results training over compressed text use EqualInfoAC with 16-bit windows and vocabulary size at either 65k (best) or 256 (second-best). One clear advantage of the $v$=65k model is that it has a $2\times$ better token compression rate, so sees twice as much raw text during training. To assess whether its performance gain is due entirely to this advantage, we train a 25m parameter M2 model over the same dataset, but reduce its sequence length from $512 \rightarrow 256$. This model trains on half as many *tokens*, but sees the same amount of underlying text as the $v$=256 model.[16]

Table 6 shows that even in this setting, the model with larger vocabulary is stronger.[17] In fact, *most* of the bits/byte gain (84% absolute) is due to the structural change in tokenization, as opposed to the additional text seen. One possible explanation for its strong performance is that the $v$=65k model uses exactly one token to represent each equal-info window. We'll see in the next section that in EqualInfoAC settings with multiple tokens per window, any non-initial tokens are highly context-dependent, and learning proceeds on a curriculum from the "easy" window-initial tokens to the "harder" window-final tokens.

## 6 Analysis

In this section we examine how neural compression based tokenizers differ from standard tokenizers, and conduct additional analysis on training dynamics and learnability of compressed data. This analysis leads us

---

[14] The slight gain is statistically insignificant ($p = 0.07$).

[15] This could also be explained by how the end of sequence is handled, see Appendix D.5

[16] To compensate for the smaller number of tokens in a sample of 20 batches from validation set when each example is 256 tokens, we compute our evaluation metrics over 40 batches.

[17] It may be possible to achieve further gains by increasing the token bit depth further. However, most deep learning frameworks do not support using unsigned data types for inputs, and the resulting large vocabulary size can cause a computational bottleneck in the final softmax layer.

Table 6: Most of the gain of increasing vocabulary from 256 to 65k remains even in the "byte matched" setting, where the models train over the same number of raw bytes. Performance gains seen between settings are all statistically significant.

| Tokenization | Comparison | Bits/Byte |
|---|---|---|
| EqualInfoAC[$b$=16, $v$=256] | | $1.472 \pm 0.004$ |
| EqualInfoAC[$b$=16, $v$=65k] | byte matched | $1.287 \pm 0.003$ |
| EqualInfoAC[$b$=16, $v$=65k] | token matched | $1.251 \pm 0.003$ |

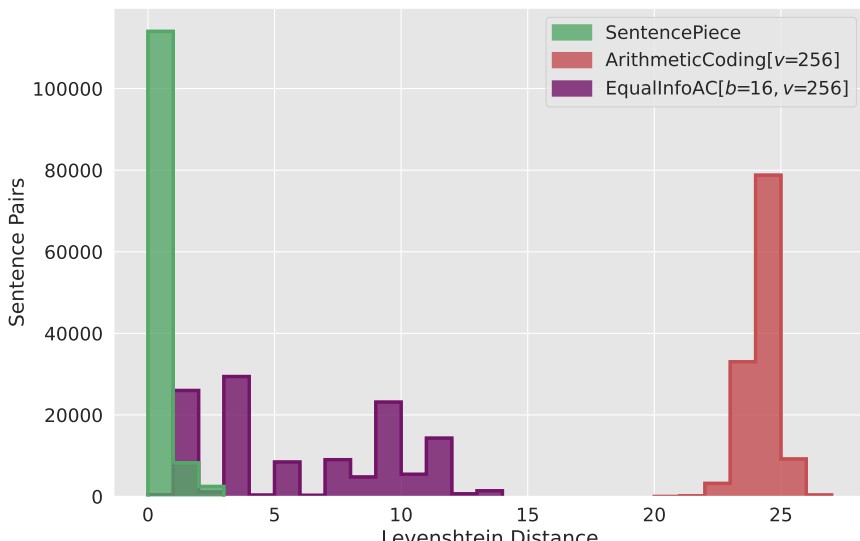

Figure 8: Edit distance between tokenized sentence pairs that differ only by a single prefix word. With SentencePiece, changing the prefix has a minimal effect on tokenization, limited to the prefix itself. AC has the largest edit distances, as the prefix affects the probabilities M1 assigns to all subsequent text. EqualInfoAC falls in the middle, as many distinct prefixes lead to the same windowing of subsequent text, and each window is encoded in isolation.

to several recommendations for future work developing new compression schemes that aim to be learnable by transformer models while delivering stronger compression than subword tokenizers.

## 6.1 AC-Based Tokenizations are Less Stable and Less Semantic than SentencePiece

We observe that text tokenized by our AC-based methods is not "stable"—that is, the tokenization of similar input data often leads to vastly different tokens sequences. To quantify this stability, we measure the effect of adding different prefixes to a fixed test sentence. We sample 500 words from a list of 3,000 common English words[18] and prepend each to the example sentence in Table 7. We then tokenize these new sentences using SentencePiece, AC[$v$=256], and EqualInfoAC[$b$=16, $v$=256]. Finally, for each tokenizer, we calculate the Levenshtein edit distance (Levenshtein, 1965) between all pairs of tokenized sentences.

Fig. 8 shows a histogram of these distances. We observe that SentencePiece tokenized sentence pairs always have a low edit distance, as the effect of changing the prefix is limited to the prefix tokens. In contrast, the edit distance between AC-tokenized sentence pairs is consistently high. The presence of the prefix affects the probabilities M1 assigns throughout the *entire* sequence, resulting in widely varying token outputs. Edit distance with EqualInfoAC falls in between these two extremes. The degree to which the tokens after the prefix "match" depends on how the prefix aligns with EqualInfo window boundaries, which in turn affects the windowing of the rest of the sentence. Since there are a relatively small number of possible window

---

[18]https://www.ef.com/ca/english-resources/english-vocabulary/top-3000-words/

Table 7: Comparing tokenization under SentencePiece vs. EqualInfoAC. SentencePiece gives a fairly stable text → token mapping. For instance, each occurrence of "`elephants`" maps to the same two-token sequence: [ `elephant`] [`s`]. By contrast, EqualInfoAC[$b$=16, $v$=65k] is less stable and less semantic. Each occurrence of "`elephants`" maps to different tokens, and most tokens fail to align with meaningful linguistic boundaries (e.g., word or morpheme).

| Input Text | The three currently living species are: African savanna elephants, African forest elephants, and the Asian elephants. |
|---|---|
| **SentencePiece Tokens** | [The] [ three] [ currently] [ living] [ species] [ are] [:] [ African] [ ] [s] [a] [v] [anna] [ elephant] [s] [,] [ African] [forest] [ elephant] [s] [,] [ and] [ the] [ Asian] [ elephant] [s] [.] |
| **EqualInfoAC [$b$=16, $v$=65k] Tokens** | [The th] [ree c] [urrently l] [iving ] [species] [ are] [: A] [frica] [n sav] [anna] [ ele] [pha] [nts, ] [Afr] [ican ] [forest ] [eleph] [ants, ] [and the ] [Asi] [an e] [lep] [hant] [s.] |

alignments, many sentence pairs have highly overlapping token sequences. As AC tokenization is so much less stable than the other settings, we restrict further analysis to SentencePiece and EqualInfoAC.

While the performance of our EqualInfoAC[$b$=16, $v$=65k] model approaches that of the SentencePiece baseline, our edit distances analysis suggests that the two tokenization schemes differ in many regards. To better understand these differences in qualitative terms, we compare the tokenizations of a single sentence in Table 7.

First, we observe that SentencePiece produces a stable text → token mapping. For example, "`elephants`" appears three times in the sentence, and stably maps to the same two-token sequence in all cases: [ `elephant`] [`s`]. Similarly, both occurrences of "`African`" map to the same token: [ `African`]. In contrast, the EqualInfoAC tokenization is relatively unstable, with each occurrence of these words being segmented in a different way, and yielding different token sequences.

Second, we find that the SentencePiece tokenization is more "semantic", by which we mean that the segmentation it induces aligns better with meaningful linguistic units—words and morphemes. While there are some exceptions, e.g. "`savanna`" being tokenized as [`s`] [`a`] [`v`] [`anna`], the more common case is that whole words are parsed as single tokens (e.g., `currently`), or into meaningful morphemes (e.g., `elephant-s`). By comparison, EqualInfoAC tokenization appears to almost entirely disregard word and morpheme boundaries. As one example, we see "`Asian elephants.`" tokenized as [`Asi`] [`an e`] [`lep`] [`hant`] [`s.`].

Despite these differences, there is an important *similarity* between SentencePiece and EqualInfoAC[$b$=16, $v$=65k]: they are both stable in the token → text direction. That is, a given token ID, e.g., token #500, will always map to the same output text. This "transparent decoding" property likely makes it easier for a downstream model to learn over these tokens.[19]

When we move to versions of EqualInfoAC that contain *multiple* tokens per window, such as EqualInfoAC[$b$=16, $v$=256], this transparency is destroyed for all non-initial tokens within a window. This is illustrated in Table 8. When the same token appears window-initially in different contexts, we see the window text has a stable prefix—e.g., token #151 always maps to the prefix "`le-`". However, when occurring as the *second* token within a two-token window, there are no apparent correspondences between window text.[20] As EqualInfoAC window length increases, the proportion of tokens that are stable decreases. This may explain the observed difficulty of learning over longer windows. The window text for all instances of these tokens can be seen in Appendix D.8.

Note that Table 8 examines window → text, as opposed to token → text correspondences. This is because for multi-token windows, the mapping from tokens to text is not well defined. More specifically, each

---

[19]Padding to reach a specific window size can require extra computation to discern between padding and characters that compress to all zeros, however we find in Appendix D.5 that it is not an issue for M2 models.

[20]A repeated text substring that happens to be aligned with a window multiple times is one of the few cases where the second token will represent the same text.

Table 8: Window-initial tokens have stable token → text mappings, while non-initial tokens have contextual meaning and are thus unstable. We tokenize 20 documents with EqualInfoAC[$b$=16, $v$=256] and show the full window text in a random sample of cases where a specific token appears at the first or second position within the window.

| Token | Window Position | Window Text |
|---|---|---|
| 151 | 1 | [lew ] / [lea] / [led] / [len] / [less] / [led] / [les] / [lew ] |
|  | 2 | [thoug] / [ust] / [ this] / [etti] / [npo] / [thoug] / [ un] / [imag] |
| 185 | 1 | [ord a] / [or k] / [ord] / [or f] / [or al] / [or a ] / [ore i] / [ora] |
|  | 2 | [ery] / [s may] / [cian] / [onte] / [h de] / [cri] / [opp] / [ides] |

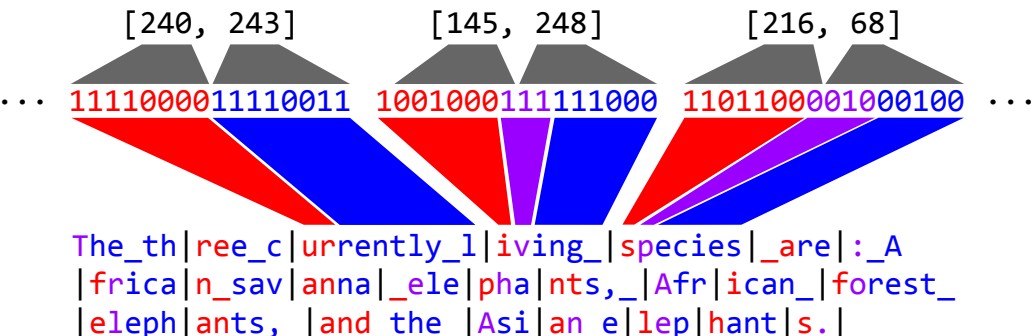

Figure 9: An illustration of the mapping between characters (bottom), bits (middle) and tokens (top) in the EqualInfoAC[$b$=16, $v$=256] setting. Each equal-info window corresponds to 16 bits of AC output, which are chunked into two 8-bit M2 tokens from a vocabulary of 256. Colors indicate whether each character contributes to the first token, second token, or both tokens within a window. We note that window-initial characters are not well compressed, so the initial 8-bit token tends to only cover one or two characters.

character maps to a particular subsequence of the compressed bitstream, but these may not align with token boundaries.[21] Fig. 9 illustrates the mapping between characters, bits, and tokens. We find that many windows contain a character (shown in purple) whose bits are split across two 8-bit tokens.

Fig. 9 also highlights that window-initial characters are not being well compressed, with the window-initial token often only covering one or two characters. This is due to our EqualInfoAC procedure fully resetting M1's context at every window boundary. With no context, M1 cannot make confident predictions, leading to more bits being needed to represent the initial character. While this hurts the overall compression ratio, there is a benefit for learnability in that each window can be decoded in isolation. In Appendix A.2, we experiment with maintaining M1 context across two or more windows, and find that the improvement in compression ratio does not justify the degradation in bits/byte.

## 6.2 AC Decoding is Learned Step-by-Step

As Arithmetic Coding is a sequential (left-to-right) and contextual algorithm, the text represented by a given token will differ based on the previous token. As such, a model should perform better on a token if it has a strong understanding of the token before it. When using EqualInfoAC compression, each window represents an independent Arithmetic Coding document. As we move deeper into the window, more and more AC decompression must be done to understand the token.

---

[21]This can be a source of instability, even in window-initial tokens, see Appendix C.

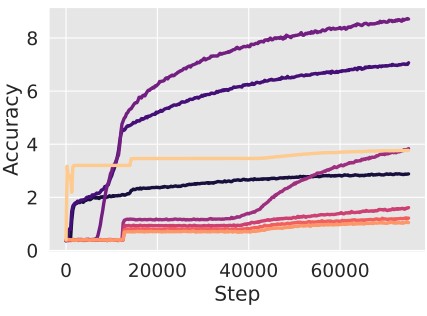

(a) Accuracy per token position

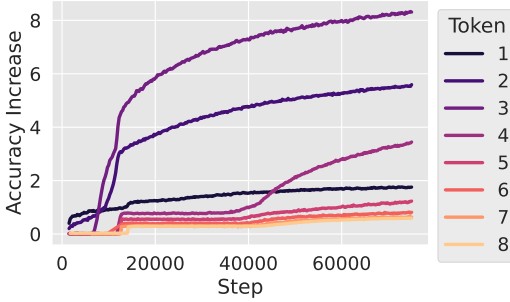

(b) Increase over "trivial" accuracy per token position

Figure 10: Earlier tokens within the 8-token window of an EqualInfoAC[$b$=64, $v$=256] model are learned earlier in training. As training progresses, the model "unlocks" the ability to model tokens deeper and deeper into the window. The plot on the right shows the increase over "trivial" accuracy—which we define as the maximum accuracy achieved in the first 2,000 steps of training. (Note, window-final padding makes trivial accuracy higher for later positions.) For tokens #1–3, later tokens reach higher accuracy ($3 > 2 > 1$), likely due to the benefit of local context. For tokens #4–8, accuracy deteriorates, indicating that the model has trouble tracking the AC algorithm for more than ~32 bits.

To understand how a token's position within a window affects learning, we track the average accuracy at each position within the 8-token windows of a 403m parameter EqualInfoAC[$b$=64, $v$=256] model[22] during training. Fig. 10 shows both raw accuracy (left) as well as the increase over "trivial" accuracy (right), which we define as the maximum accuracy achieved in the first 2,000 steps of training. By that point in training, the model predictions no longer change drastically between updates, and trivial patterns, such as how end-of-window padding generally reduces the number of possible tokens at the ends of windows, have already been learned. Looking at accuracy increase highlights the "sequential learning" trend by discounting any part of accuracy that is text independent. In particular, we note that window-final tokens have a non-uniform distribution due to the use of window-final padding bits (see our EqualInfoAC formulation in Section 3.3), which can be learned without any understanding of the text.

We observe two interesting trends. First, there is a clear ordering as to when the model starts to make meaningful (non-trivial) progress on a given position. The initial token (#1) is learned first, followed fairly quickly by #2 and then #3. Later tokens are only "unlocked" after 10,000 training steps, suggesting that the ability to model these tokens builds on a foundation of understanding the preceding tokens within the window.

The second trend concerns the accuracy reached at each position. Here, we observe an increase in accuracy from #1 < #2 < #3, followed by a decrease from #3 < #4 < #5 and so on.[23] We interpret the increase across the first three positions as due to the benefit of extra leftward context.[24] This is akin to the initial byte in a word being harder to predict than the following bytes. The decreasing performance at tokens #4 and beyond suggests the model is unable to track AC decompression indefinitely. While the model clearly learns to decompress longer sequences as training progresses, reliably decoding past 32 bits of AC output appears to be a challenge.

## 6.3 Learnable Distributions are Less Uniform

A well-known result in the compression literature is that there can be no recursive compression (Mahoney, 2013). The compression algorithm removes information captured by its model, resulting in a uniform output

---

[22]The absolute accuracy of the EqualInfoAC[$b$=64, $v$=256] model is relatively poor, but its relatively long window provides the clearest illustration of these positional trends. We observe similar trends for EqualInfoAC[$b$=16, $v$=256] which has smaller windows but much stronger performance.

[23]The final token #8 also fits this trend when looking at the increase over non-trivial accuracy. The raw accuracy is higher than previous tokens, #4–7, due to the skewed distribution induced by window-final padding.

[24]While suggests that 24 bit windows could be effective, but in Appendix A.1 we found it is not.

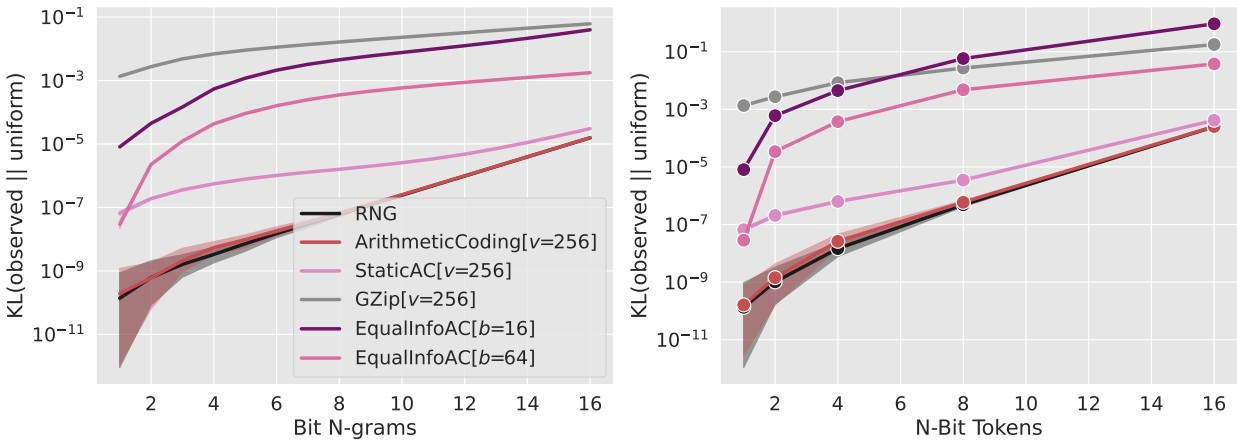

(a) *bit* n-grams counting all overlapping occurrences      (b) n-bit *tokens* following our M2 tokenization

Figure 11: As the bitstream is grouped into larger units, the empirical distribution moves away from uniform. We plot KL divergence of observed n-gram distributions from the uniform distribution, across various n-gram sizes. While AC compressed data would be difficult to distinguish from random data, we find there are still patterns to capture when using other compression schemes, particularly for GZip and shorter EqualInfoAC windows. Compared to the left plot, we find that the tokenized bitstream (see Section 3.4) has even more information for M2 to capture.

that appears random to the original model. However, our setting is not recursive compression. Instead, a separate and larger model is trained on the compressed output, which should be able to capture new patterns in the bitstream.

Despite this, the output of compression using M1 appears very uniform, as evidenced by the minimal gains from modeling the unigram token distribution in Table 3. Therefore, it seems reasonable that this uniformity could make it hard for M2 to learn (as all patterns must be contextual). We investigate this by plotting the KL divergence (Kullback & Leibler, 1951) between the observed empirical distribution and a uniform distribution for different segmentations of the bitstream. If the underlying distribution of bits was truly random and independent, then the distribution of unigrams for some bitstream segmentation should remain uniform as $p(b_i, \ldots, b_{i+n}) = \prod_{j=i}^{i+n}(p(b_j))$ and therefore the KL divergence should remain close to zero. On the other hand, if the distribution diverges from uniform, there is contextual information to be learned when training an LLM to model $p(b_n|b_i, \ldots, b_{i+n-1})$.

We segment the bitstream either into *bit n-grams*, where successive n-grams are allowed to overlap, or into *n-bit tokens*, following our M2 tokenization procedure—see Section 3.4. We only plot tokenization into *n*-bits that are factors of 16, otherwise tokens would cross window boundaries in the EqualInfoAC[*b*=16] setting.

As a baseline, we used the cryptographic `secrets` package in Python to generate bitstreams that should be truly random and independent. As such, the KL divergence should remain at 0 when segmented in the same way as the compressed data. The reason this does not hold in Fig. 11 is that the maximum likelihood estimate of entropy, $\hat{H} = -\sum_{x \in \hat{\mathcal{X}}} \hat{p}(x) \log_2 \hat{p}(x)$, is negatively biased (Paninski, 2003). In Fig. 17 we see that when using a Miller-Madow estimator (Miller, 1955) to correct for this bias, the expected KL of 0 is well within sampling noise bounds. To account for noise in the entropy estimation, we plot 90th percentile intervals of the KL divergence between the observed entropy from 100 disjoint samples of the data and the uniform distribution.[25]

---

[25]As the number of bits in a segmentation grow, the vocabulary size increases exponentially, requiring many more samples. Thus we expect noise in the entropy estimate to grow with *n*. This holds, but it is obfuscated by the log scaling in Fig. 11. In fact, the magnitude of the noise for settings such as GZip and EqualInfoAC is larger than for AC or RNG. This noise behavior is seen in Fig. 16. See Appendix D.6 for more information on entropy estimation and bias correction.

The AC and RNG lines in Fig. 11 are very similar and their sampling noise intervals have large overlaps. This suggests that the data generated by AC compression with M1 is difficult to distinguish from random data.[26] This is a possible explanation for why M2 models trained on AC data only learn to output a uniform distribution, as seen in Fig. 3.

In Fig. 11, we see that GZip is the least uniform, which is expected as it has the worst compression rate among these settings. However, the segmentation into tokens does not result in much extra information. This is again suggestive that the differences between the "coding" components of GZip and Arithmetic Coding are important for learnability. It is also a possible explanation of why GZip is the one setting where using 16-bit tokens does not improve performance.

Similarly, Fig. 11 shows that EqualInfoAC[$b$=16] has the most information among the Arithmetic Coding approaches. Given that this is the most learnable setting, it suggests that non-uniformity of the bitstream may be important for learning. We also see a large increase when moving to 16-bit tokens, providing a further possible explanation for why larger vocabulary is helpful (see Section 5.3). Finally, we note that StaticAC has less information than EqualInfoAC[$b$=16], suggesting that weakening the "coding" component of Arithmetic Coding is a more effective way to retain information and increase learnability for M2.

## 7 Conclusion

We have shown there is promise in the idea of training LLMs over neurally compressed text. In the best case, this will allow training over text that is better compressed than standard subword token sequences, while maintaining learnability. This an appealing prospect, as models that read and write more text per token are more efficient to train and serve, and can model longer dependencies.

While the "very simplest" approach does not work (training directly over a tokenized AC-encoded bitstream), we showed that a relatively simple modification—compression via Equal Info Windows—already brings us within striking distance of popular tokenizers. When measured in terms of perplexity achievable at fixed inference cost (FLOPs/byte), we find that our method outperforms raw byte-level models, and comes increasingly close to the performance of SentencePiece tokenization as scale increases to 2 billion parameters.

While bespoke compression methods have developed around different modalities (e.g., text, audio, images, video) and different applications (e.g., delta-of-delta for regular repeating timestamps (Pelkonen et al., 2015)), to our knowledge, no efficient compression methods have been designed specifically for use as LLM tokenizers. We are optimistic that future work will create such methods. Compared to today's subword tokenizers, we expect these methods (i) will deliver higher compression rates, (ii) will come closer to equal information per token, thus allocating compute more effectively, and (iii) will give models a more direct view of the underlying raw text, thus helping on spelling and pronunciation tasks. As a tradeoff, we expect these neural tokenizers will be *somewhat* less stable in their text ↔ token mapping, but perhaps not so unstable as our approach here. In particular, we think it is worth exploring methods under which a given word typically maps to a relatively small number (tens not thousands) of relatable token sequences.

One direction we left unexplored is the idea of passing information between the compressing model (M1) and the LLM trained over compressed text (M2). Some additional signal of M1's internal state or output may be helpful for M2 to accurately simulate M1, which is a prerequisite to flawlessly encoding and decoding M1-compressed text.

For hill-climbing in this space, we found it useful to iterate on the sequence-to-sequence sub-tasks of compression and decompression, which should, in theory, be learnable with high accuracy. Specifically, if future work can devise a strong (~10×) compressor that a transformer can be trained to accurately encode and decode, we expect that this will be an ideal candidate for tokenizing text for LLMs.

---

[26]For $n > 2$, the AC entropy is statistically significantly less than the RNG entropy, however, differences in the mean entropy only start to appear after ~8 decimal places.

**Acknowledgments**

We would like to thank Ben Adlam, Grégoire Delétang, Rosanne Liu, and Colin Raffel for detailed comments on an earlier draft. We're also grateful to Peter Liu, both for helpful discussion, as well as for building some of the infrastructure that made our experiments easier to run. Finally, we thank Doug Eck, Noah Fiedel, and the PAGI team for ongoing guidance and feedback.

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

## A  Additional Experiments

### A.1  Equal-Info Windows with 24 bits

In Fig. 10, we saw that when trained on 64-bit Equal-Info windows, our model achieves the highest accuracy on the third 8-bit token. All else being equal, we expect to see higher accuracy on later tokens, since the model should benefit from additional leftward context. Thus, the stark accuracy decrease from token #3 (>8%) to token #4 (<4%) likely indicates a difficulty in tracking the AC algorithm beyond 24 bits.

In this section, we explore training over 24-bit windows. First, we train a 403m M2 model over EqualInfoAC[$b$=24, $v$=256] compressed data, following the procedure from Section 3. We find this model achieves 1.394 bits/byte. This fits cleanly in the trend seen in Fig. 5—while the model outperforms EqualInfoAC[$b$=32, $v$=256] (which includes the problematic token #4), it still underperforms EqualInfoAC[$b$=16, $v$=256].

We also train a model using EqualInfoAC[$b$=24, $v$=65k], and find it achieves 1.249 bits/byte, again fitting cleanly between the 16-bit and 32-bit settings. It is noteworthy that the model performs well despite the fact that the token bit-depth (16) is not a divisor of the window size (24). This results in M2 tokens that cross window boundaries.

EqualInfoAC[$b$=24, $v$=256] has a compression ratio of 3.15 and EqualInfoAC[$b$=24, $v$=65k] has a compression ratio of 6.30.

## A.2 Resetting M1 Every Window is Beneficial

We saw in Section 6.1 that the initial characters within each equal-info window are especially costly to encode. This is due to our procedure of resetting the M1 model's context at every window boundary. In this section, we experiment with maintaining M1 context over multiple windows. We expect this to improve the overall compression rate, but to negatively impact learnability, as a window can no longer be decoded in isolation.

As one extreme case, we consider *never* resetting M1. That is, we first use M1 to assign probabilities to all tokens in the input, and then use Equal-Info AC to compress the sequence into multiple fixed sized windows. In this setting we find that M2 does not learn beyond outputting a uniform distribution over tokens.[27]

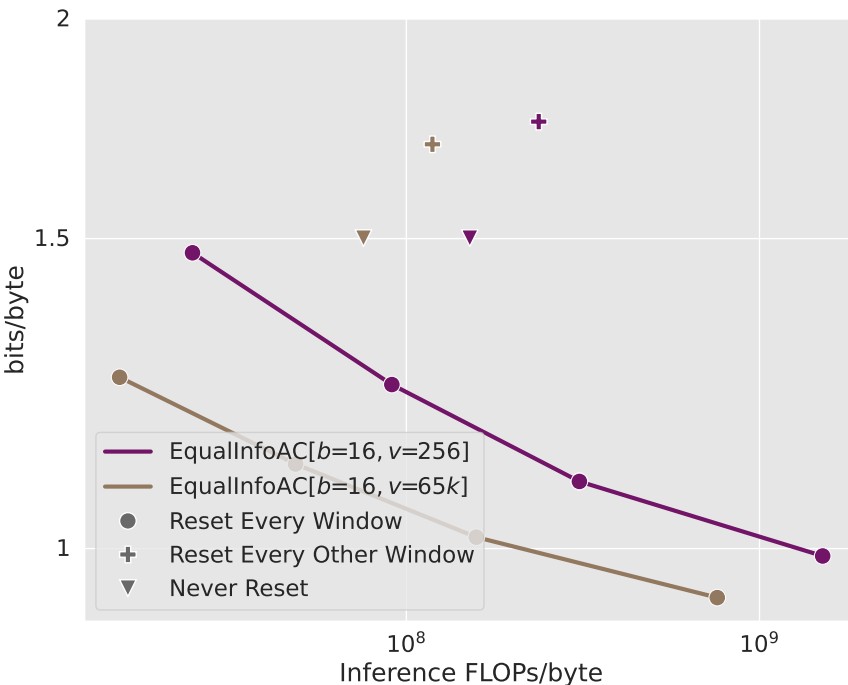

Figure 12: Resetting M1 with each Equal-Info window is beneficial. While reducing how often M1 is reset improves the compression ratio (see Table 9), bits/byte performance is much worse, as seen by these points being far above the scaling curves for our default settings.

At the other extreme, we consider resetting M1 every *other* time an Equal-Info window is emitted. We train 403m parameter M2 models in each of these settings, and show the results in Fig. 12 and Table 9. While resetting M1 every other window does not yield as strong performance as our standard EqualInfoAC[$b$=16, $v$=65k] setting, it is notable that it is still learning at all. This suggests that M2 is able to leverage context from compressed windows earlier in the sequence. While the improved compression ratio places this compression scheme further to the right in Fig. 4 than EqualInfoAC[$b$=16, $v$=65k] or SentencePiece, the reduced performance means M2 must be extremely large to reach the Pareto frontier.

Exploring this compression scheme in a more traditional pre-training setting, where M2 is trained on trillions of tokens, would be interesting as this model seems particularly under-fit. Similarly, exploring how many windows can be emitted before resetting M1 while remaining learnable for M2 would be of interest as future work.

---

[27]Consequently, models trained on compressed data that used 8-bit or 16-bit tokenization have the same performance in terms of bits/byte. 16-bit tokenization squares the number of possible symbols in the vocabulary. This results in a doubling of the negative log likelihood loss when predicting the uniform distribution. This effectively cancels out the doubling of the compression rate 16-bit tokenization yields.

| Dataset | M1 Reset | Compression Ratio | bits/byte |
|---|---|---|---|
| EqualInfoAC[$b$=16, $v$=256] | every window | 2.66 | 1.092 |
| | every other window | 3.40 | 1.748 |
| | never | 5.33 | 1.501 |
| EqualInfoAC[$b$=16, $v$=65k] | every window | 5.31 | 1.015 |
| | every other window | 6.80 | 1.697 |
| | never | 10.66 | 1.501 |

Table 9: Resetting M1 less often yields greater compression rate, but bits/byte performance degrades. We see that resetting M1 every other window results in more compressed, but still learnable data. However, M2 models trained on that data do not perform as well as when M1 is reset every Equal-Info window. As seen in Fig. 3 never resetting M1 seems to have strong performance, but this is an artifact of its high compression rate; it does not actually learn anything beyond a uniform distribution.

### A.3 Character Awareness and Spelling

Language models trained over subword tokens are not "character-aware"—that is, they lack direct access to the character-level makeup of their input text. As a result, these models tend to perform poorly on tasks relating to spelling (Xue et al., 2022; Liu et al., 2023). In this section, we explore whether our M2 models trained over neurally compressed text can offer an improvement in this regard.

We investigate spelling ability using an adapted version of the WikiSpell task (Liu et al., 2023). When fed a word as input, the model is required to spell it by outputing the component characters as separate tokens. We compare two of the 403m parameter models trained in Section 3—the EqualInfoAC[$b$=16, $v$=256] M2 model and the SentencePiece model. In each case, we fine-tune the model (which has been pre-trained for 200,000 steps on language modeling) on the WikiSpell task for 200 steps—5 epochs over the data. We continue the same learning rate schedule as pre-training.

For each model, we tokenize the WikiSpell input words with the same tokenizer used during pre-training. The choice of which token IDs to use for the *output* character sequence is less straightforward. For the M2 model, we use the ByT5 vocabulary (Xue et al., 2022), which overlaps in an arbitrary way with the 256 IDs used during pre-training. For the SentencePiece model, we consider two options. For a more direct comparison, we map each output character to an arbitrary ID shared with a non-character subword token in the vocabulary—"SentencePiece (Shared)". As an alternative, we also consider mapping each output character to the ID of that character within the SentencePiece vocabulary—"SentencePiece (Characters)". We expect this setting to perform better, as the model should already have some understanding of these characters and their relation to larger subwords from pre-training.

Fig. 13 shows that a SentencePiece model leveraging its pre-trained character representations has the best spelling performance on held-out evaluation samples. It is also the first that reaches 100% training accuracy. The EqualInfoAC[$b$=16, $v$=256] model, which must share token meanings between the compressed input and the character output, outperforms a SentencePiece model with the same limitation on the evaluation set. Additionally, the model pre-trained over compressed text reaches 100% training accuracy before the analogous SentencePiece model. Interestingly, M2's inability to reach 100% accuracy suggests that the model may not actually be performing decompression internally while doing language modeling over compressed text.[28]

### A.4 GZip Headers and Footers

GZip compressed documents have both a header—two bytes that identify the file type—and a footer—two bytes representing the Adler-32 checksum (Deutsch & Gailly, 1996) of the input. We train 25m M2 models on versions of the dataset where the header/footer are removed, as well as versions where it is kept. In

---

[28]We note that only 0.18% of WikiSpell dev examples (9 of 5,000) contain a character that is unseen during training. Thus, even if the output character IDs are chosen arbitrarily, a character-aware model should in theory be able to get nearly perfect performance.

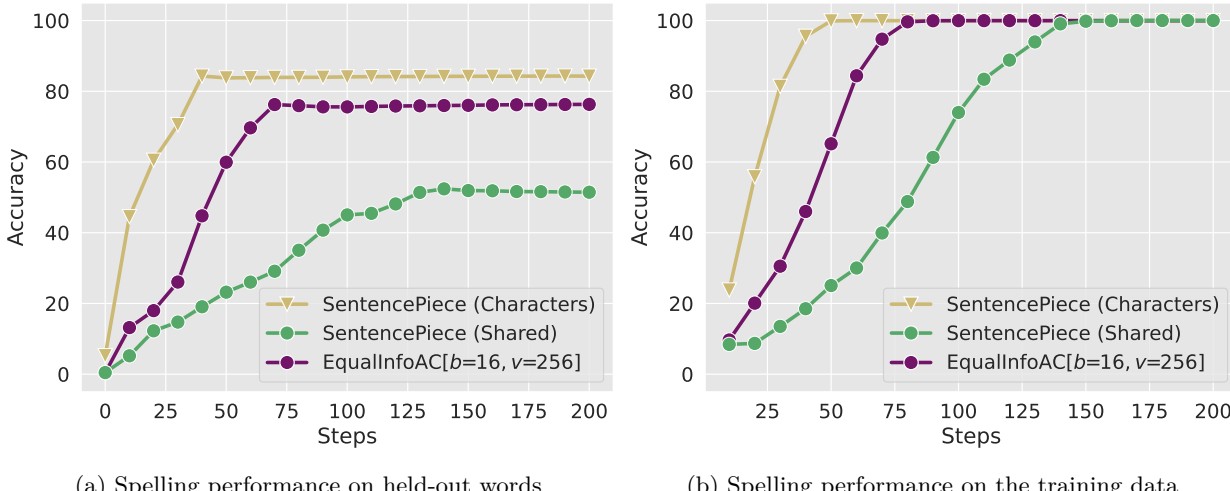

(a) Spelling performance on held-out words

(b) Spelling performance on the training data

Figure 13: Reusing SentencePiece character representations yields the strongest speller (and the fastest to train), but when forced to learn the output representations from scratch, a pre-trained EqualInfoAC[$b$=16, $v$=256] model is a better speller than a SentencePiece based model.

Table 10, we see that including the header/footer offers a marginal improvement in bits/byte, at the cost of a marginal decrease in compression ratio. These differences are not large enough to overcome GZip's poor performance compared to other compression methods. We opt to use versions of GZip including the header/footer throughout.

Table 10: Removal of the GZip header and footer results in minimal performance differences.

| Method | Compression Ratio | bits/byte |
|---|---|---|
| GZip[$v$=256] | 2.23 | 2.33 |
| −header/footer | 2.24 | 2.35 |
| GZip[$v$=65k] | 4.46 | 2.91 |
| −header/footer | 4.47 | 2.92 |

### A.5 Avoiding End-of-Window Zeros

Our Equal-Info Windows algorithm described in Section 3.3 involves compressing text until *exceeding* a specified number of bits (e.g., 16-bits per window), then backtracking one character, and padding with zero bits. As a character may encode to all zero bits, we need to ensure there is no ambiguity between padding bits and all-zero characters in order to achieve lossless compression. Our default solution, used in all experiments, is to greedily include the most characters possible in each window. Given the knowledge of greedy encoding, we can design a decoder that decodes windows unambiguously by using look-ahead, as detailed in Appendix D.5.

An alternative resolution to the problem of window-final all-zero characters is to simply avoid emitting these characters during compression—instead, delaying their encoding until the next window. While this degrades the compression rate, it results in a much simpler decoding algorithm. Specifically, with the knowledge that a window includes the *least* number of characters that can result in the given bitstream, we can decode windows without any look-ahead.

To test whether this alternative scheme (Delay) improves learnability, we retrain EqualInfo M2 models following the procedure from Section 3, with the only difference being that in generating M2's training data, we delay emission of window-final all-zero characters. Applying this "delayed" version of windowed compression, we observe a reduction in compression rate from 2.66 to 2.20.

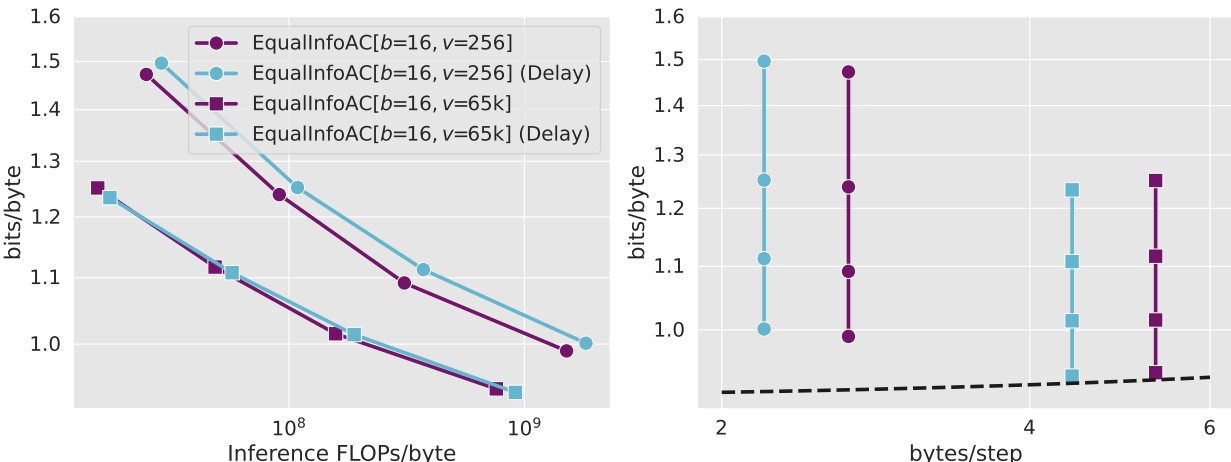

Figure 14: Comparison of our default "greedy" implementation of EqualInfoAC with a variant (Delay) where emission of window-final all-zero characters is delayed. For EqualInfoAC[$b$=16, $v$=256], the default implementation performs better. For EqualInfoAC[$b$=16, $v$=65k], bits/byte performs improves slightly but at the cost of compression rate.

Fig. 14 shows the results of training M2 models over data compressed with each method. For EqualInfoAC[$b$=16, $v$=256], we find that our default implementation outperforms the delayed version along both axes. For EqualInfoAC[$b$=16, $v$=65k], the delayed implementation delivers a slight improvement in terms of bits/byte. However, this gain is offset by the reduction in compression ratio, so the two performance curves largely overlap. Numerical values can be found in Table 17. Overall, our findings suggest that when trained over data encoded with our default (greedy) EqualInfoAC algorithm, M2 is able to distinguish trailing zeros that represent characters from those that represent padding. We opt to use our default implementation throughout, as it maximizes compression ratio.

## B   Other Compression Methods Considered

### B.1   Equal-Text Windows

"Equal-Text" windows are a simpler alternative to Equal-Info windows. Rather than consuming a variable amount of text and outputting a fixed number of bits, Equal-Text windows feed a fixed amount of text into the Arithmetic Coder, which compresses to a variable number of bits.

We anticipate a downside to Equal-Text windows is that it would be difficult for M2 to decipher where one window ends and another begins, as this no longer aligns with a fixed number of tokens, and the boundary may appear token-internally. We considered adding delimiter tokens between windows to overcome this difficulty. However, we expect this would degrade the compression rate too much, especially for the short AC compressed windows that we found most effective in Fig. 5.

Further exploration of Equal-Text windows, especially to see if the delimiters are actually required, would be interesting future work as the Equal-Text Windows algorithm is much simpler than Equal-Info Windows.

### B.2   Huffman Coding

We also considered using Huffman Coding (Huffman, 1952) as a baseline compression implementation. However, as most implementations use static probabilities for characters, the resulting compression rate would likely be too low to be competitive. With static Huffman Coding, there is a fixed mapping between bitstream subsequences and characters, which may improve learnability by M2 models. However, because the

coding component assigns each character a whole number of bits, the coding is less less optimal compared to Arithmetic Coding.

Huffman Coding can be made adaptive by updating the induced codebook periodically, based on newer data. When considering bit-level compression, adaptive Huffman Coding performs similar to static Huffman Coding (Mahoney, 2013). However, when considering token-level compression, and the fact that the adaptive distribution will come from M1, not unigrams of the recent data, training M2 models on adaptive Huffman Coding could be interesting future work. As Huffman coding is part of the GZip algorithm, we opted not to explore Huffman Coding on its own.

### B.3   Asymmetric Numeral Systems

Another compression algorithm we considered was Asymmetric Numeral Systems (ANS) (Duda, 2014). ANS has strong coding performance and is amenable to adaptive probabilities. The internal state is only a single natural number, which may be easier for an LLM to track than the two real numbers used in Arithmetic Coding (AC). However, unlike AC, the encoding and decoding algorithm are stack-like, where the encoder runs left-to-right and the decoder runs right-to-left. We thought this would make streaming inference, where a single M2 token is generated and then decoded by M1 before another M2 token is generates, difficult. Thus we opted to explore AC over ANS in this work. However, the simpler state is appealing and using ANS for compression would be of interest as future work.

## C   Instability in the Initial Token of Multi-Token Windows

There are cases where the token $\rightarrow$ text mapping for the initial token in a multi-token EqualInfoAC window can be unstable. When a character's bitstream crosses the token boundary—the purple characters in Fig. 9—only some prefix of the bitstream contributes to the value of the initial token. It is possible that another character may produce a different bitstream with a shared prefix. If the token boundary comes *before* the difference in the bitstreams, then the two tokens will have the same value but represent different text. When this occurs the text prefix will remain stable, i.e., any characters whose bitstreams are entirely contained within the initial token will match, but the final character may differ. Thus the notion of mapping a compressed token to exact characters is not well defined, as there are often cases there a character is spread across two tokens. Note, this only occurs at token boundaries; EqualInfoAC[$b$=16, $v$=65k] is stable as no characters cross windows. This is most likely a reason that EqualInfoAC[$b$=16, $v$=65k] outperforms EqualInfoAC[$b$=16, $v$=256] in our byte-controlled ablations (Section 5.3). Therefore, we consider EqualInfoAC stable enough to enable learnability by M2.

Interestingly, Külekci (2011) point out this same issue, where a fixed size view of a variable length stream can cause false equivalencies when prefixes match. Similar to our findings, they find the models do have some limited ability to deal with these situations.

## D   Details for Reproducibility

### D.1   Variance

Sampling from the validation set was seeded. For a given seed, the same batches are sampled at each evaluation step within a training run. Similarly, when models of a different size are trained on the same compressed data, the same evaluation batches are sampled, allowing for fair comparison. As the Bytes and SentencePiece baselines use deterministic datasets, the validation seed is not used. Instead the "start_step" is incremented by 20 to get a new sample of 20 batches.

Model initialization and the order of the training data is controlled by the training seed. This seed was also changed during variance testing. During training, the dataset is checkpointed and therefore each example is seen exactly once. The exact order of the training data is determined by the seed. As the Bytes and SentencePiece baselines use deterministic datasets, the training order is fixed.

Table 11: Variance in performance is low. Even with maximum changes between runs—different evaluation samples, different training orders, and different parameter initialization—there is very little variance in final performance. Statistics were calculated over 5 different 25m parameter training runs for each method.

| Method | bits/byte |
|---|---|
| Bytes | $1.2899 \pm 0.0020$ |
| SentencePiece | $1.1171 \pm 0.0006$ |
| AC[$v$=256] | $1.4573 \pm 0.0001$ |
| StaticAC[$v$=256] | $4.6936 \pm 0.0005$ |
| EqualInfoAC[$b$=16, $v$=256] | $1.4724 \pm 0.0044$ |
| EqualInfoAC[$b$=32, $v$=256] | $2.0457 \pm 0.0058$ |
| EqualInfoAC[$b$=64, $v$=256] | $1.8494 \pm 0.0052$ |
| EqualInfoAC[$b$=128, $v$=256] | $1.7121 \pm 0.0003$ |
| GZip[$v$=256] | $2.3374 \pm 0.0061$ |

5 models with 25m parameters were trained with different seeds (both validation and training) for each compression method and the two baselines. The mean and standard deviation can be found in Table 11. The variance is so low that we only report single values for most other experimental settings, such as larger models.

Training models of size 403m and 2b over data compressed with EqualInfoAC[$b$=64, $v$=256] and EqualInfoAC[$b$=128, $v$=256], as well as a 2b model with EqualInfoAC[$b$=128, $v$=65k], occasionally diverged, collapsing to a simple model that just output the uniform distribution. The numbers for these settings exclude these divergent runs. This resulted in 7 re-runs in the most problematic case.

### D.2 The Amount of Raw Text Bytes Seen by M2

Table 12 shows the number of tokens and bytes found in the training dataset for each compression method. During the data generation process, sequences of 10,240—generated by concatenating 128 C4 byte-tokenized documents together—are compressed. Some of these sequences, namely the final sequence created from the tail of the concatenated docs, are too short to be compressed to the target length of 512. Thus, the exact number of tokens in the dataset can vary slightly. With no padding, each dataset would have been trained on 26,214,400,000 tokens, we see all settings are close to this value, with the maximum deviation being EqualInfoAC[$b$=128, $v$=65k] with 1.06% fewer tokens. All compression datasets are created from the same source sequences, thus the underlying byte sequences compressed by weaker methods are prefixes of the underlying sequences compressed by stronger methods.

### D.3 Scaling Curves with Scaled Training Data

When scaling models, Hoffmann et al. (2022) argue that training data should be scaled linearly with model size. As such, when comparing settings with constant training FLOPs, a large part of the FLOPs budget should be used by adding more training data. We apply this technique to compensate for our 2b models being under-trained by plotting the scaling curves in Fig. 15, where the smaller models are trained with *less* data, proportional to their size. Models with 25m parameters only train for 3k steps, 113m for 11k, 403m for 40k, and 2b for 200k steps. Otherwise, the settings match those in Fig. 3. Numerical values used in the graph can be found in Table 13.

Scaling the training data adjusts the absolute slopes of the lines for all models that learn. Models that do not learn still only predict a uniform distribution. The trends between settings are unchanged. Thus we opt to plot the versions where training data is held constant across model sizes.

Table 12: Compression ratios (bytes / tokens) achieved by various methods. For EqualInfoAC, the compression ratio increases with increased window size. Small differences in the number of non-padding tokens are due to noise in the data generation process—some randomly selected documents are too short to be compressed to the target length of 512.

| Method | Compression Ratio | Tokens | Bytes |
|---|---|---|---|
| Bytes | 1.0 | 26,188,185,600 | 26,188,185,600 |
| SentencePiece | 4.28 | 26,112,163,840 | 111,728,726,639 |
| AC[$v$=256] | 5.49 | 26,083,328,000 | 143,197,470,720 |
| StaticAC[$v$=256] | 1.73 | 26,175,078,400 | 45,282,885,632 |
| GZip[$v$=256] | 2.23 | 26,175,209,472 | 58,370,424,832 |
| EqualInfoAC[$b$=16, $v$=256] | 2.66 | 26,154,106,880 | 69,569,924,301 |
| EqualInfoAC[$b$=32, $v$=256] | 3.49 | 26,109,542,400 | 91,122,302,976 |
| EqualInfoAC[$b$=64, $v$=256] | 4.16 | 26,110,853,120 | 108,621,148,979 |
| EqualInfoAC[$b$=128, $v$=256] | 4.61 | 26,078,085,120 | 120,219,972,403 |
| AC[$v$=65k] | 10.98 | 25,952,256,000 | 284,955,770,880 |
| StaticAC[$v$=65k] | 3.46 | 26,133,135,360 | 90,420,648,346 |
| GZip[$v$=65k] | 4.47 | 26,122,649,600 | 116,768,243,712 |
| EqualInfoAC[$b$=16, $v$=65k] | 5.31 | 26,091,192,320 | 138,544,231,219 |
| EqualInfoAC[$b$=32, $v$=65k] | 6.97 | 26,049,249,280 | 181,563,267,482 |
| EqualInfoAC[$b$=64, $v$=65k] | 8.33 | 26,004,684,800 | 216,619,024,384 |
| EqualInfoAC[$b$=128, $v$=65k] | 9.22 | 25,936,527,360 | 239,134,782,259 |

### D.4 Evaluation Details

In our experiments, different settings have different vocabulary size, tokenization, and has a different amount of underlying text due to variations in compression rate. Thus, they are not directly comparable using "per-token" versions metrics like the cross-entropy, negative log likelihood loss, or perplexity. To address this, we convert our token-level negative log likelihood loss, $\ell$, to byte-level negative log likelihood loss by dividing the loss by that compression method's specific token-level compression rate, $\ell_{\text{byte}} = \ell/(L_{iT}/L_{oT}) = \ell(L_{oT}/L_{iT})$. Note that we use "per byte" metrics over "per character" metrics as there is ambiguity as to what counts as a character when working with UTF-8 Unicode.

As is common in evaluation of work related to compression, instead of the negative log likelihood loss $\ell_{\text{byte}}$ (in the unit of "nats") per byte, we use bits/byte. This would require using log base two instead of the natural log during the negative log likelihood calculation, but this conversion can be done after the fact, bits/byte $= \log_2(e^{\ell_{\text{byte}}}) = \ell_{\text{byte}}/\ln(2)$. Note that this results in the same conversion used in Gao et al. (2020), bits/byte $= \ell_{\text{byte}}/\ln(2) = (L_{oT}/L_{iT})\ell/\ln(2)$, when the input tokens represent bytes.

As one of the main advantages of an M2 model that processes compressed text is that it needs to be run over fewer tokens, we also compare models based on the amount of FLOPs required during inference. Different compression methods result in different sequence lengths for the M2 model to process. Therefore, we need to standardize our FLOPs measurement to the byte-level so that it is comparable across methods. We start with FLOPs/token—approximated by $2 \times$ num_params (not including embedding parameters) following Kaplan et al. (2020)—and divide it by that method's token-level compression rate to get the FLOPs/byte, just like the bits/byte conversion. For methods that require running an M1 model over each byte, the FLOPs/byte cost of the M1 model is added. Note, while there is a computational cost to running GZip over the input text, we ignore it as it is insubstantial compared to the cost of running model inference.

Evaluation of language models is often done by running the model on the entire validation set, moving the sliding window formed by the model's context window by a single token at each step. This yields stronger models by providing the most context possible when making predictions for a token. As we care about relative performances between methods, opposed to absolute performance, we opt to evaluate the model on a sample of the C4 validation set. During evaluation, the model is run over 20 batches, resulting in predictions

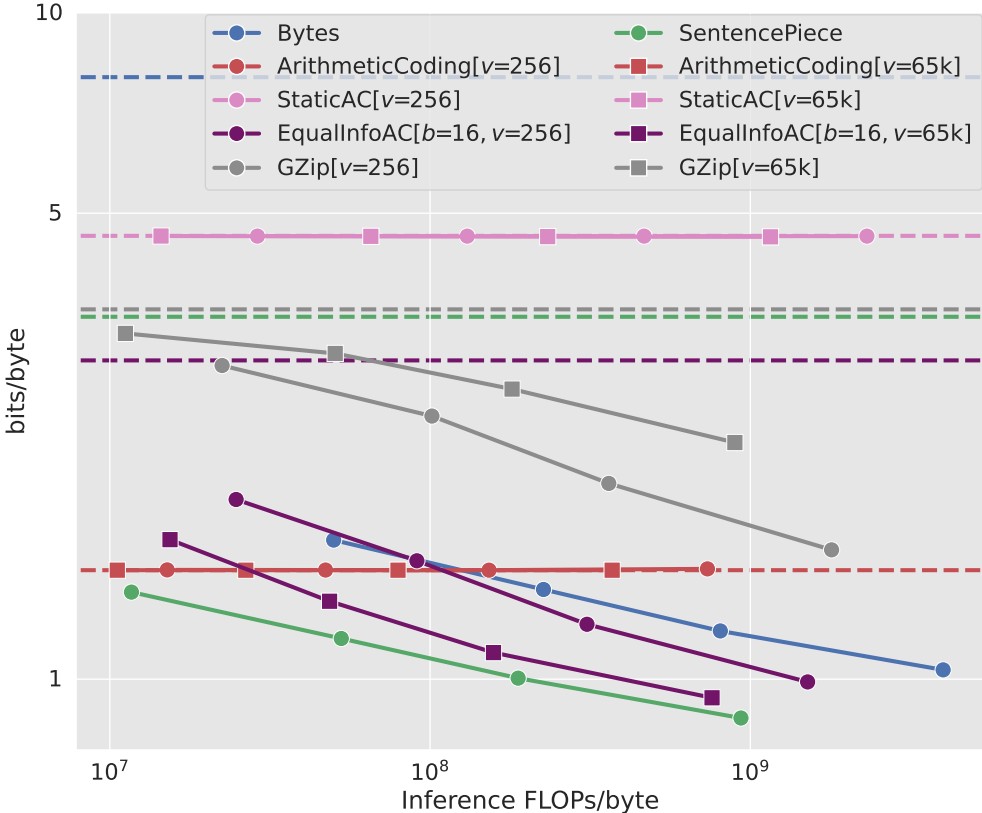

Figure 15: Training language models over compressed text while scaling training data with model size results in steeper slopes. When scaling model size, it has been found that the training data should be scaled proportionally (Hoffmann et al., 2022). We apply this scaling technique by plotting values for smaller models at earlier training steps. The trends are similar to Fig. 3, even down to things like where the EqualInfoAC[$b$=16, $v$=256] line crosses the Bytes baseline (between the 25m and 113m parameter models).

for 2,621,440 tokens. These tokens represent different amounts of text based on the compression method, thus it would have been impossible to run evaluation on the same bytes for all methods. We trained five 25m parameter models with different seeds and found that the final performance is very stable. The largest standard deviation was 0.0061. Thus, the variance introduced from sampling the validation set is negligible. See Appendix D.1 for more information.

## D.5 End of Window and End of Input Sequence Handling

In the implementation of EqualInfoAC[$b$=$W$], each output window must end up being $W$ bits. Therefore, when the compression of an additional character would result in a bitstream of more than $W$ bits, padding of the compressed bitstream *without* that additional character must be done.

In cases where the final character in the window only adds zeros to the bitstream, it is unclear at first glance if that final character was included in the window, or if it was omitted and the trailing zeros are all padding. However, the compression scheme is still lossless if we are consistent in our encoding. By *always including the most input characters possible in each window*, we know that, during decoding, if the addition of a final character (which is compressed to all zeros) still results in the same compressed bitstream, then that final character is part of that window. The decoding algorithm also knows when to stop adding characters to input—when the addition of a new character would generate more than $W$ bits when compressed.[29]

---

[29]The only exception is at the end of the sequence, when the amount of padding is dictated by running out of input characters instead of running out of room in the window. This can be solved by including an end-of-input symbol.

Table 13: Numerical values from Fig. 15. Values for the uniform distribution and FLOPs/byte values can be found in Table 15.

| Dataset | Size | Step | bits/byte |
|---|---|---|---|
| Bytes | 25m | 3k | 1.62 |
| | 113m | 11k | 1.36 |
| | 403m | 40k | 1.18 |
| | 2b | 200k | 1.03 |
| SentencePiece | 25m | 3k | 1.35 |
| | 113m | 11k | 1.15 |
| | 403m | 40k | 1.00 |
| | 2b | 200k | 0.87 |
| AC[$v$=256] | 25m | 3k | 1.46 |
| | 113m | 11k | 1.46 |
| | 403m | 40k | 1.46 |
| | 2b | 200k | 1.46 |
| AC[$v$=65k] | 25m | 3k | 1.46 |
| | 113m | 11k | 1.46 |
| | 403m | 40k | 1.46 |
| | 2b | 200k | 1.46 |
| StaticAC[$v$=256] | 25m | 3k | 4.62 |
| | 113m | 11k | 4.62 |
| | 403m | 40k | 4.62 |
| | 2b | 200k | 4.62 |
| StaticAC[$v$=65k] | 25m | 3k | 4.62 |
| | 113m | 11k | 4.62 |
| | 403m | 40k | 4.61 |
| | 2b | 200k | 4.61 |
| EqualInfoAC[$b$=16, $v$=256] | 25m | 3k | 1.86 |
| | 113m | 11k | 1.50 |
| | 403m | 40k | 1.21 |
| | 2b | 200k | 0.99 |
| EqualInfoAC[$b$=16, $v$=65k] | 25m | 3k | 1.62 |
| | 113m | 11k | 1.31 |
| | 403m | 40k | 1.10 |
| | 2b | 200k | 0.94 |
| GZip[$v$=256] | 25m | 3k | 2.95 |
| | 113m | 11k | 2.48 |
| | 403m | 40k | 1.97 |
| | 2b | 200k | 1.56 |
| GZip[$v$=65k] | 25m | 3k | 3.30 |
| | 113m | 11k | 3.08 |
| | 403m | 40k | 2.72 |
| | 2b | 200k | 2.26 |

This kind of padding is present in many Arithmetic Coding implementations and is generally solved by either giving the AC decoder the original input sequence length and the compressed message, or by the AC decoder using a special termination character. These fixes are hard to apply in our setting. Passing the number of tokens present in a window to M2 would be possible during training, but it would make inference much more complex (requiring a solution such as M2 generating "fertility" scores that specify how many characters the generated tokens represent (Brown et al., 1993)). In order to include an end-of-input symbol for the AC decoder, the M1 model must be able to assign reasonable probabilities to that symbol. Therefore it would need to appear at the end of each training example, hindering M1's ability to be applied to longer sequences.

As such, we achieve lossless compression within each window by allowing the AC decoder to be run multiple times, incrementing the sequence length until we find the sequence that, when compressed, no longer matches the compressed output and backtracking. As we do not include an AC decoder end-of-input symbol, when we reach the end of the input, there is some ambiguity in the number of characters that are included in that window. However, as we compress extremely long sequences, over 10,000 characters, the final window in a, trimmed, M2 example rarely correlates to the final input characters. Thus most sequences (99.4%) are decompressable using the approach above. In our validation data, just 0.0011% of windows represent the end of an input sequence. Additionally, the standard deviations of the loss across validation tokens is similar for models trained over compressed input and models trained directly on SentencePiece. Thus we believe that these ambiguous tokens do not effect our results.

## D.6    Entropy Estimation

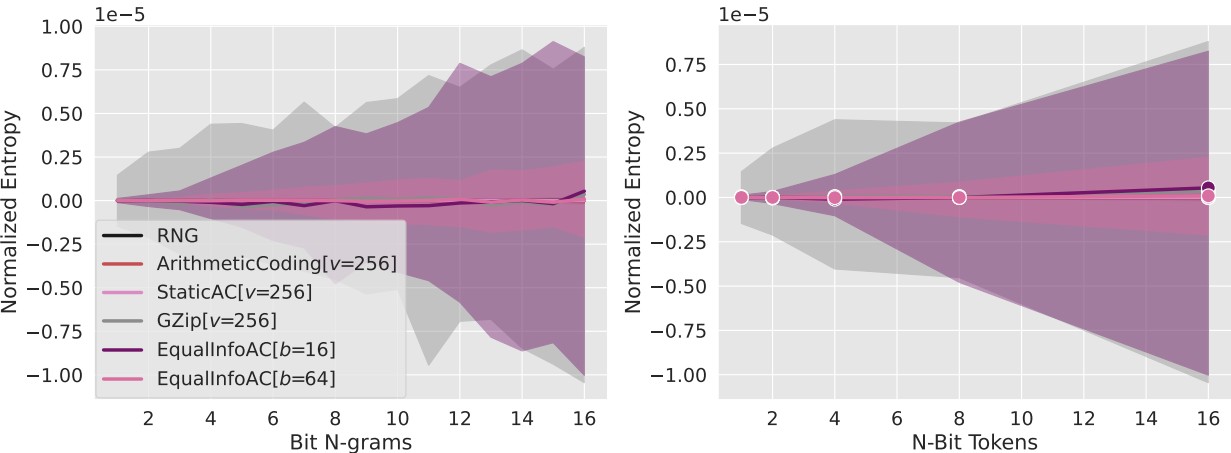

(a) *bit* n-grams counting all overlapping occurrences      (b) n-bit *tokens* following our M2 tokenization

Figure 16: The amount of noise in the entropy estimate grows as the length of bit segments grow. Larger segmentations of the bitstream result in larger vocabularies and therefore require larger sample sizes for accurate entropy estimates. For each setting, we plot the 5%, 50%, and 95% percentile intervals for the entropy, normalized by the average entropy across partitions. We see that the noise grows with $n$ and that settings like EqualInfoAC[$b$=16] are noisier than AC, despite this not being apparent in Fig. 11.

To account for noise in the entropy estimation, we partition the data into 100 disjoint samples. This results in each partition being a sample of ~2 billion symbols for n-grams and ~130 million for tokens. We then calculate the entropy for each partition and the KL divergence between the entropy of the 0.5, 0.50, and 0.95 quantile points and a uniform distribution. These quantiles are then plotted on Fig. 11 to illustrate sampling noise—90% of sampled entropies fall within these bounds. The log scaling of Fig. 11 hides some of the noise trends, namely that the noise grows with $n$ and that settings like GZip and EqualInfoAC are noisier than AC and RNG. These trends are seen in Fig. 16 where the entropy has been normalized based on the mean entropy calculated across the partitions.

The maximum likelihood, or plug-in, estimator of entropy, $\hat{H} = -\sum_{x \in \mathcal{X}} \hat{p}(x) \log_2 \hat{p}(x)$, is negatively biased—in fact, all entropy estimators are biased (Paninski, 2003). The Miller-Madow estimator attempts to correct for this bias by adding the approximate bias, caused by sampling, to the plug-in estimator.[30] The Miller-Madow estimator is given by $\hat{H}_{MM} = \hat{H} + \frac{|\hat{V}|-1}{2m}$. In this case, $m$ is the size of the sample used to estimate entropy and $|\hat{V}|$ is the estimated vocabulary size. In some applications, the vocabulary may need to be estimated—for

---

[30]There are other methods for entropy bias correction such as those used in DeDeo et al. (2013) based on bootstrapping (Efron, 1979), however, with the size of the C4 training data, the required resampling was not possible. Thus, we use Miller-Madow in this work.

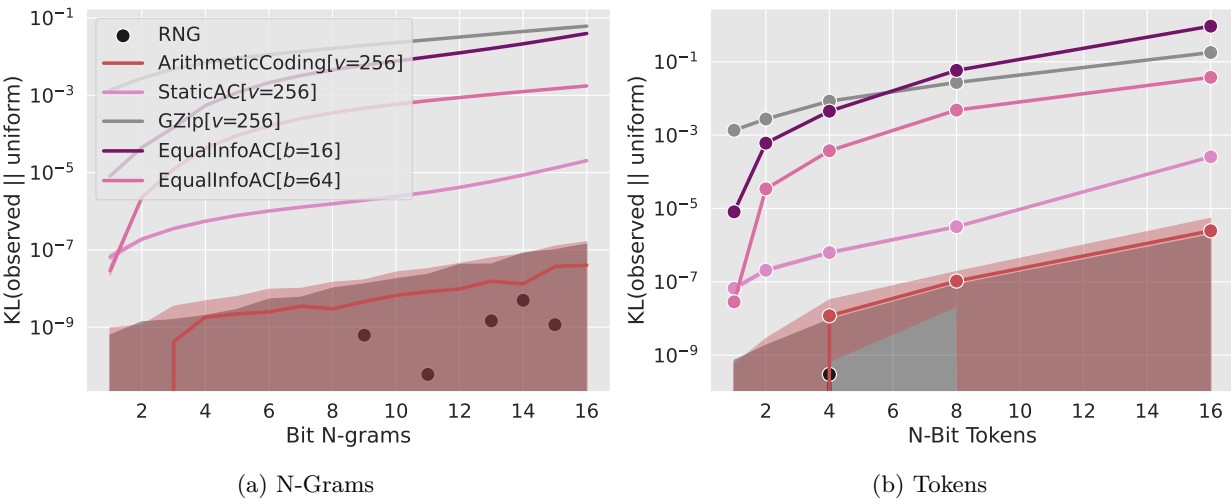

Figure 17: Bias corrected KL divergence between the observed and uniform distributions for different segmentations of the bitstream. This plot is similar to Fig. 11, however, the KL divergence calculations use the entropy of the observed distribution after applying the Miller-Madow bias correction. After applying bias correction, we see that the expected 0 KL divergence for the RNG baseline is now within the 90th percentile bounds. However, this can results in an, incorrect, negative KL divergence which is removed from the graph. Thus the RNG 50th percentile is shown as a scatter plot rather than a broken line. In this setting it is clear that the 50th percentile for $AC[v=65k]$s above the 50th percentile for RNG, however, it is hard to disentangle the two as their 5th percentile lines are similar.

example, to account for new words that are added to languages—but in this case our vocabulary size is always $2^n$ where $n$ is the size of the current segmentation.

When we plot the KL divergence between the Miller-Madow estimated entropy and the uniform distribution, we see that the percentile interval for the RNG baseline now includes 0, the KL divergence we expect given the data was generated from random and independent bits. As bias correction is approximate, it is possible that, for a given sample, the correction will result in an entropy greater than the maximum entropy possible for a given vocabulary size. Given that KL divergence between a distribution $P$ and the uniform distribution $U$ simplifies to the entropy of $U$ minus the entropy of $P$, $\text{KL}(P||U) = H[U] - \hat{H}[P] = \log_2|V| - \hat{H}[p]$, this results in a negative KL divergence, which is not allowed. These points get removed from the graph during log scaling and the resulting 50% percentile line for RNG data looks strange. Therefore, we only plot points with positive KL divergence in Fig. 17. The Miller-Madow estimation of entropy makes it clear that the 0.5 entropy quantile for AC compressed data is much higher than the 50% percentile for RNG data. Additionally, for $n > 2$, the AC entropy is statistically significantly less than the RNG entropy; however, differences in the mean entropy only start to appear after ~8 decimal places. This slight difference in mean, coupled with the fact that the 5% percentiles are similar, means we cannot confidently assert the model will be able to easily distinguish the AC compressed data from random data. Given that we care about the differences between the entropy of data compressed with different methods—which is invariant to bias—and the strange plots when values are less than 0, we opt to plot the plug-in estimator in Fig. 11 instead of the Miller-Madow estimator.

## D.7 Analysis Implementation

Matplolib (Hunter, 2007) and Seaborn (Waskom, 2021) were used to make all the included graphs.

Statistical significance tests were done using Welch's t-test (Welch, 1947) using the function `scipy.stats.ttest_ind_from_stats` from SciPy (Virtanen et al., 2020). We used $p < 0.05$ as the statistical significance threshold.

### D.8 Window Text Patterns and Token Positions

We tokenize 20 documents of length 1,024 with EqualInfoAC[$b$=16, $v$=256] and find that all 256 possible token values occur multiple times, both as the first and as the second token within the window. When tokenized with EqualInfoAC[$b$=16, $v$=65k], 34.5% of attested tokens appear more than once. Table 14 shows all the window text for repeated tokens.

Table 14: The deduplicated window text from all instances of tokens that appear multiple times when we tokenized 20 documents of length 1,024 (20,480 compressed tokens) with EqualInfoAC[$b$=16, $v$=256].

| Token | Window Position | Window Text |
|---|---|---|
| 185 | 1 | [or ] / [or a ] / [or ac] / [or al] / [or cr] / [or d] / [or f] / [or h] [or hi] / [or i] / [or k] / [or ma] / [or pr] / [or r] / [or s] / [or se] [or su] / [or t] / [or to] / [or v] / [or wha] / [or y] / [or yo] / [or, t] [or-] / [or.] / [ora] / [orc] / [orce ] / [ord] / [ord a] / [order] [ore a] / [ore e] / [ore ev] / [ore g] / [ore i] |
|  | 2 | [ 4] / [ of F] / [ records ] / [. Lo] / [Alt] / [OI] / [ase ] / [at y] [cian] / [cri] / [d. I] / [ery] / [h de] / [hen s] / [ides] / [n ne] [oft] / [om i] / [onte] / [opp] / [pir] / [rev] / [reve] / [s may] [tion a] / [y do] / [y t] |
| 151 | 1 | [le] / [le s] / [le t] / [le. ] / [lea] / [lec] / [led] / [led ] [led t] / [leg] / [lege] / [leh] / [lem ] / [leme] / [lems] / [len] [ler] / [les] / [less] / [let] / [lett] / [level] / [lew ] / [ley] / [lf ] |
|  | 2 | [ all ] / [ nut] / [ this] / [ un] / [. I w] / [Ni] / [as t] / [ceed ] [choos] / [e Mi] / [e-li] / [etti] / [imag] / [ion a] / [k a] / [ne a] [ng up] / [niversi] / [npo] / [nt pr] / [pi] / [rvices] / [s T] / [s your] [s?] / [so c] / [stag] / [thou] / [thoug] / [ust] / [ust ] |

### D.9 Numerical Values

Table 15 includes the specific values used to create Fig. 3. Similarly, Table 16 includes the values used to create Fig. 5. The numerical values from Fig. 6 can be found across Table 15 and Table 16. Table 17 includes the numerical values from Fig. 14.

Table 15: Numerical values from Fig. 3. Methods that use 16-bit tokens ($v$=65k) have the same uniform distribution performance as the 8-bit version ($v$=265). Note: One thousand million is used over one billion to make comparison of FLOPs/byte values easier.

| Dataset | Size | bits/byte | FLOPs/byte |
|---|---|---|---|
| Bytes | 25m | 1.29 | 50.00M |
| | 113m | 1.16 | 226.00M |
| | 403m | 1.08 | 806.00M |
| | 2b | 1.03 | 4,000.00M |
| | uniform | 8.00 | - |
| SentencePiece | 25m | 1.12 | 11.69M |
| | 113m | 1.01 | 52.82M |
| | 403m | 0.94 | 188.37M |
| | 2b | 0.87 | 934.84M |
| | uniform | 3.47 | - |
| AC[$v$=256] | 25m | 1.46 | 15.11M |
| | 113m | 1.46 | 47.17M |
| | 403m | 1.46 | 152.81M |
| | 2b | 1.46 | 734.60M |
| | uniform | 1.46 | - |
| AC[$v$=65k] | 25m | 1.46 | 10.55M |
| | 113m | 1.46 | 26.58M |
| | 403m | 1.46 | 79.41M |
| | 2b | 1.46 | 370.30M |
| StaticAC[$v$=256] | 25m | 4.61 | 28.90M |
| | 113m | 4.61 | 130.64M |
| | 403m | 4.61 | 465.90M |
| | 2b | 4.61 | 2,310.00M |
| | uniform | 4.62 | - |
| StaticAC[$v$=65k] | 25m | 4.60 | 14.45M |
| | 113m | 4.60 | 65.32M |
| | 403m | 4.62 | 232.95M |
| | 2b | 4.62 | 1,160.00M |
| EqualInfoAC[$b$=16, $v$=256] | 25m | 1.47 | 24.80M |
| | 113m | 1.23 | 90.96M |
| | 403m | 1.09 | 309.01M |
| | 2b | 0.99 | 1,510.00M |
| | uniform | 3.01 | - |
| EqualInfoAC[$b$=16, $v$=65k] | 25m | 1.25 | 15.42M |
| | 113m | 1.12 | 48.56M |
| | 403m | 1.02 | 157.79M |
| | 2b | 0.94 | 759.30M |
| GZip[$v$=256] | 25m | 2.34 | 22.42M |
| | 113m | 1.93 | 101.35M |
| | 403m | 1.69 | 361.43M |
| | 2b | 1.56 | 1,790.00M |
| | uniform | 3.59 | - |
| GZip[$v$=65k] | 25m | 2.92 | 11.19M |
| | 113m | 2.69 | 50.56M |
| | 403m | 2.48 | 180.31M |
| | 2b | 2.26 | 894.85M |

Table 16: Numerical values from Fig. 5. Values for EqualInfoAC[$b$=16, $v$=256] and EqualInfoAC[$b$=16, $v$=65k] can be found in Table 15. Note, EqualInfoAC[$b$=128, $v$=256] showed slight improvements beyond the significant digits shown here as the model scales.

| Dataset | Size | bits/byte | FLOPs/byte |
|---|---|---|---|
| EqualInfoAC[$b$=32, $v$=256] | 25m | 2.05 | 20.33M |
| | 113m | 1.94 | 70.76M |
| | 403m | 1.83 | 236.95M |
| | 2b | 1.65 | 1,150.00M |
| EqualInfoAC[$b$=32, $v$=65k] | 25m | 1.95 | 13.17M |
| | 113m | 1.85 | 38.42M |
| | 403m | 1.74 | 121.64M |
| | 2b | 1.63 | 579.89M |
| EqualInfoAC[$b$=64, $v$=256] | 25m | 1.85 | 18.02M |
| | 113m | 1.82 | 60.33M |
| | 403m | 1.80 | 199.75M |
| | 2b | 1.79 | 967.54M |
| EqualInfoAC[$b$=64, $v$=65k] | 25m | 1.82 | 12.00M |
| | 113m | 1.80 | 33.13M |
| | 403m | 1.79 | 102.76M |
| | 2b | 1.76 | 486.19M |
| EqualInfoAC[$b$=128, $v$=256] | 25m | 1.71 | 16.85M |
| | 113m | 1.71 | 55.02M |
| | 403m | 1.71 | 180.84M |
| | 2b | 1.71 | 873.68M |
| EqualInfoAC[$b$=128, $v$=65k] | 25m | 1.70 | 11.42M |
| | 113m | 1.69 | 30.51M |
| | 403m | 1.68 | 93.42M |
| | 2b | 1.67 | 439.84M |

Table 17: Numerical values from Fig. 14, comparing our standard implementation of EqualInfoAC versus an implementation where window-final all-zero symbols are delayed.

| Dataset | Compression Ratio | Size | bits/byte | FLOPs/byte |
|---|---|---|---|---|
| EqualInfoAC[$b$=16, $v$=256] | 2.66 | 25m | 1.47 | 24.80M |
| | | 113m | 1.23 | 90.96M |
| | | 403m | 1.09 | 309.01M |
| | | 2b | 0.99 | 1,510.00M |
| EqualInfoAC[$b$=16, $v$=256] Delay | 2.20 | 25m | 1.50 | 28.73M |
| | | 113m | 1.25 | 108.73M |
| | | 403m | 1.11 | 372.36M |
| | | 2b | 1.00 | 1,820.00M |
| EqualInfoAC[$b$=16, $v$=65k] | 5.31 | 25m | 1.25 | 15.42M |
| | | 113m | 1.12 | 48.56M |
| | | 403m | 1.02 | 157.79M |
| | | 2b | 0.94 | 789.30M |
| EqualInfoAC[$b$=16, $v$=65k] Delay | 4.40 | 25m | 1.23 | 17.36M |
| | | 113m | 1.11 | 57.36M |
| | | 403m | 1.01 | 190.18M |
| | | 2b | 0.93 | 915.09M |

