# OpenReview forum: "Training LLMs over Neurally Compressed Text"
_TMLR — Accepted by TMLR_

### Review · Reviewer_nNhF · 2024-09-23

**Summary Of Contributions:**

### Main exploration idea
The authors explore the effects of modifying tokenization in LLM training by replacing standard methods with alternative compression techniques. This problem is particularly intriguing due to the potential benefits of higher compression, such as increased context length during training and inference, along with reduced latency. However, the challenge is amplified by the authors’ unique, practically motivated setup, where the probability model used to compress the raw text is not shared with the downstream LLM, which remains unchanged. As a result, the LLM must either learn to decode the compressed input or directly extract meaningful patterns from it. Given that compression reduces statistical correlations, this makes the task of learning from the compressed tokens even more difficult.

###  Experimental Setup summary
In particular, authors extensively study arithmetic coding (`AC`) for compressing the text. The experimental setup is to use a “weak” LLM (context-aware transformer, `M1`) as the probability model for arithmetic compression, followed by training another “stronger” LLM (another transformer, `M2`) on the tokenized compressed bytes. The performance and efficiency of the trained M2 model is compared across M1 compressors of various sizes as well as GZIP-compressed bitstream. The approach is also compared to standard byte-level tokenizer and SentencePiece (sub-word level tokenizer). The authors study three different metrics:

- Perplexity metrics as a result of training M2 over tokens extracted from compressed text, measured in log-likelihood loss scaled for input bytes instead of tokens and converted to bits instead of nats, referred to as `bits/bytes`. This metric controls for the learning performance of the downstream LLM over compressed data.
- Computation required to perform inference over a given length of raw text, referred to as `FLOPs/byte`. This metric provides insights into the computational efficiency gains of compressing the text: benefits corresponding to being able to process higher context length.
- Number of bytes per text processed in one auto-regressive step, referred to as `bytes/step`. This metric mainly corresponds to the fact that higher compression implies fewer autoregressive steps during model decoding and therefore accounts for possible benefits for latency during serving.

###  Main findings
The main findings of the paper can be summarized as:

1. There is a tradeoff in the level of compression one might desire for optimal efficiency gains in terms of FLOPs/byte given a model performance in terms of bits/bytes.
    1. Authors observe that naively using Arithmetic coding or gzip (DEFLATE) does not lead to this optimal tradeoff.
    2. SentencePiece (sub-word) level tokenizers are optimal over the range of experimentation carried out by authors.
    3. Suitable modified Arithmetic Coding, referred to as `EqualInfoAC` can beat byte-level tokenizers.
2. Context-aware compression introduces two main challenges for the downstream M2 LLM for decoding the tokens obtained from compressed bit-stream — (i) Need to model the probability distributions as these are not shared from M1 to M2, and (ii) Need to model the context as now the same words/text might result in very different byte-stream given the past context, and hence the M2 model has to deal with the variability in input tokens during the learning phase. The authors show throw various ablations and experiments that modeling this variable context-generated tokens is extremely challenging for LLMs.
3. Authors suggest a simple modified AC scheme, called `EqualInfoAC` which truncates AC-coded bitstream to a fixed bit-width, creating independently decompressible bit-streams with finite context size. This scheme reduces the compression rate, however, it enables the learning of these tokens via the downstream M2 model.
4. Authors provide interesting analysis probing further what contributes vs what hurts the learnability of the downstream model, which seem to suggest challenges with increasing context length during compression as well as variability in the tokens generated because of variable context.

**Audience:**

Yes

**Broader Impact Concerns:**

Not applicable.

**Claims And Evidence:**

Yes

**Requested Changes:**

###  Major
- “Shorter sequence lengths require fewer autoregressive generation steps, and reduce latency”:
    - This statement obviously depends on “per-step” computational latency as well.  One way to estimate this could be taking into account `inference FLOPs/byte` instead of `bytes/step`, if inference FLOPs were to be considered a measure of latency. Figure 3 already looks at it. Figure 4 sheds light on `bits/byte` vs `bytes/step` tradeoff and indeed seems to suggest that at certain bits/byte the claimed technical statement in the corresponding section is indeed true, however, it does not justify that we can get lower inference latency at iso-performance for a given task of equal input length.
    - Similarly, in section “Equal-Info Windows make AC learnable” in Results, it is unclear where “all our SentencePiece models require 23% longer sequences to encode the same text” statement comes from. Is it just the compression ratio comparison between SentencePiece versus the compared compression scheme?
- “Section 5.2 Transformers struggle to learn Arithmetic Coding”
    - I think this experiment is very interesting and important to the major conclusions coming from the paper, but not presented clearly. I got the gist but was unable to follow the exact experimentation setup. I would strongly recommend re-writing this section with clear experimentation setup. In particular, inputs and outputs for each of the tasks — “compression”, “decompression”, and “Byte Level LM”, contrasted against the standard setup in the paper, should be clearly presented.
- One of the main claims of the paper is that it `EqualInfoAC` beats the byte-level tokenizers. However, only one byte-level tokenizer is considered in the experiments (called `Bytes` referring to `ByT5` from Xue et al., 22). It would make the paper stronger if atleast one or few other byte level standard tokenizers are compared, e.g. WordPiece or just simple Byte-pair encoding.
- `EqualInfoAC` is not exactly lossless compression scheme as explained in detail by authors in Section D.5. I agree with the author’s analysis that this does not affect any of the conclusions, however, it will still be useful to highlight this fact in the main text. Apologies if this already exists in the paper but atleast one reader missed it.

###  Minor
Technical questions and/or suggestions, in no-particular order:
- Which algorithm is used for SentencePiece — BPE or unigram? Section 3.6 is missing this detail.
- Results Section “Equal-Info Windows make AC learnable“ states: “Using 16-bit tokens (65k vocabulary) increases performance further. EqualInfoAC[b=16, v=65k] outperforms the Bytes baseline at all model sizes. It underperforms the SentencePiece baseline, but the gap diminishes.” The closing gap between the two can be presented clearly. A log-plot would be useful to highlight this difference as it seems to be the case that we have standard diminishing returns in the shown plot.
- Similarly, the above section can be written more clearly to clarify point 1 in the Major comment 1 above.
- Overall results seem to suggest that being able to eliminate the token variability is extremely important for learning. For instance, the best performing EqualInfoAC is with the set of hyper-parameters when “the model uses exactly one token to represent each equal-info window”. Can authors shed some more light on this observation? This seems to suggest to me that beating standard tokenizers under the considered setup would always be hard as any good non-IID, lossless, compression scheme typically has two parts: prediction and entropy coding. AC, or related variants such as ANS, are known to perform extremely well entropy coding however, in most real-world data, predicting non-IID alphabets results in better compression which is mostly achieved by better predictions coming from exploiting the contexts.
- In section 3.3: “with a precision of 14. The range encoding implementation uses integers with precision + 2 bits. This is enough to encode 16-bit float logits, so should not cause numerical issues as our models are trained using bfloat16.”: bfloat and fp16 have different bits assigned to exponent and hence even this case can indeed lead to numerical issues. I think this statement should be toned down.
- I think Figure 7 is particularly illustrative in terms of the differences in approach. If it can be promoted-up before the results section, it can clarify a lot of the experimentation section for the reader.
- In Figure 8: why is trivial accuracy defined as “maximum accuracy achieved in the first 2,000 steps of training.” In particular, in left-subplot, why does higher token position achieves a higher accuracy faster than lower token positions? Footnote 22 is unclear.

**Strengths And Weaknesses:**

###  Strengths
- The problem is well-formulated, interesting and timely.
- The authors conduct extensive empirical experimentations and ablations supporting their findings throughout the paper.
- The authors provide detailed justification for various possible corner cases in their experimental setup, such as effect of over-flowing or under-flowing during the finite-window truncation in their proposed scheme. They also try to explain differences with the related work when the findings are contradictory.
- Details for reproducing the experiments are clearly provided.
- For ML/LLM readers, the compression schemes are sufficiently introduced in the paper.
- Results are presented clearly with experiments presented in a section and summarized in the title heading of the corresponding section.

###  Weaknesses
 Summary of weaknesses is provided, for details please look at requested changes.
- The paper is unclear in certain sections and the writing and/or presentation can be improved.
- The conclusion around reduced latency might be true, but is not well supported via empirical findings like the rest of the results.
- I wonder if authors could release the open-source code to their project, especially given the time spent in meticulously describing each experiment in the paper.

---

### Review · Reviewer_Kqn7 · 2024-10-03

**Summary Of Contributions:**

Summary: This is a good paper on the intersection between two topics: language modelling and compression. The motivation is that modern LLMs often train with tokenization to deal with the long sequence context/generation. From this point, the authors take a step further and combine the training with neural compression technique, hoping to achieve further compression rate and better generalization. While on the surface this idea looks simple, in fact it is very tricky since the the bit stream produced from the neural compressor can make it difficult for the language model to learn the prediction. As far as I understand, for example, using a naive arithmetic coding, a character 'a' can be mapped into two different bit-level representation, depends on the previous context and thus make it difficult for LLM to learn the representation. Furthermore, the bitstream also appears very randomly, as expected from the AC point of view, which makes the learning challenging.

The paper addressed this problem well by introducing a “Equal-Info Windows” method, i.e. divides the bitstream into series of N bits window, to address this problem and show competitive results compared to the current dictionary-based tokenization methods. While this may be a suboptimal as each window is compressed independently, I think this is a good direction towards this problem.

**Audience:**

Yes

**Broader Impact Concerns:**

This paper provides a good direction for sequence compression and modelling. No ethical concerns from my side.

**Claims And Evidence:**

Yes

**Requested Changes:**

1) This is a very good paper. I suggest the author to put the mathematical formulation/setup more concretely in the paper for future work development (both theory and practice).

**Strengths And Weaknesses:**

Strength:
1) Address an important and interesting problem of compression & language modelling. I expect future works would extend on this paper for implementation on low resource devices.
2) Develop a technique to address the challenge of learning on bit stream.
3) Results are benchmarked and studied carefullly.

Weakness:
1) The paper can be a bit more concrete problem formulation. The components between input raw text and bit-stream is not formulated mathematically for future work development.

2) (Suggestion Only) Some toy sequences, such as Markov chain [1,2], can perhaps help the reader understand more about the importance of context window N to the performance of the LLMs.

[1] Rajaraman, N., Jiao, J. and Ramchandran, K., 2024. Toward a Theory of Tokenization in LLMs. arXiv preprint arXiv:2404.08335.

[2] Makkuva, A.V., Bondaschi, M., Girish, A., Nagle, A., Jaggi, M., Kim, H. and Gastpar, M., 2024. Attention with markov: A framework for principled analysis of transformers via markov chains. arXiv preprint arXiv:2402.04161.

---

### Review · Reviewer_BCNv · 2024-11-01

**Summary Of Contributions:**

The paper introduces method to train Large Language Models (LLMs) on highly compressed text using neural compression techniques, aiming to improve efficiency by reducing sequence lengths and computational cost. Key contributions include:

1. **Neural Compression Methods**: The paper explores using **Arithmetic Coding** in conjunction with a neural language model. This neural model assigns dynamic, context-aware probabilities to each byte of text, allowing Arithmetic Coding to achieve higher compression rates. The integration of neural networks transforms the compression into an adaptive process, hence the term "neural compression." The authors analyze the challenges, such as difficulty in learning and numerical stability issues, that arise from this approach.
2. **Equal-Info AC**: A novel approach that segments text into fixed-length bit blocks, improving learnability by resetting the AC compression algorithm and model context at each window.
3. **Efficiency Gains**: Models trained on Equal-Info AC achieve shorter sequences, enhancing inference speed while underperforming in perplexity compared to traditional subword tokenizers.
4. **Interesting Result**: Extensive ablation studies show that neural compression methods, while efficient, still require improvements in stability and semantic alignment.
5. **Possible Extensions**: The paper suggests enhancements to compression-based tokenizers and calls for exploring alternative coding methods to boost learnability and performance.

**Audience:**

Yes

**Claims And Evidence:**

Yes

**Requested Changes:**

1. The hypothesis regarding the impact of near-optimal compression by Arithmetic Coding (AC) on learnability is a pivotal part of the work. If the hypothesis is that AC removes too much predictable information, making M2's learning difficult, this needs to be clearly and explicitly stated. Additionally, contextualize this point with prior works like [A] which claim that transformers are universal approximators of sequence-to-sequence functions. If this hypothesis is incorrect or incomplete, the authors should provide a clearer explanation of why M2 struggles with learning from AC-compressed data.
2. Provide additional analysis of the semantic and stability weaknesses of tokens produced by EqualInfoAC by compare these directly with standard Arithmetic Coding tokenizations. It would be good to explore how differences seen here might relate to learnability. Also, if possible, can it be contextualized with other works like [B].

[A] Are Transformers universal approximators of sequence-to-sequence functions?, ICLR 2020
[B] Getting the most out of your tokenizer for pre-training and domain adaptation, ICML 2024

**Strengths And Weaknesses:**

### Strengths

1. **Innovative Approach**: The paper introduces a novel use of neural compression - addressing efficiency in training and inference for LLMs. This integration highlights a unique direction for optimizing LLM performance.
2. **Efficiency Gains**: By compressing text into shorter sequences, the proposed method shows significant improvements in inference speed and reduces computational costs, which is crucial for deploying LLMs in resource-constrained environments.
3. **Equal-Info AC**: The introduction of Equal-Info AC is a good contribution that makes neural compression learnable. This technique ensures manageable bit-length segments and simplifies decoding, enhancing the model's ability to learn from compressed data.
4. **Interesting Results**: Several interesting questions were asked and answered ranging from "Can LLMs learn effectively from text compressed using Arithmetic Coding?" to "Do scaling laws persist?"

### Weaknesses

1. **Perplexity Trade-offs**: While the method improves efficiency, the performance in terms of perplexity still lags behind traditional subword-based models.
2. **Learnability Challenges**: The paper proposes a hypothesis which I understood to be: *"The near-optimal compression of AC means that most predictable information has been removed, making it challenging for M2 to learn anything useful."* If this is true, this is a very crucial point and needs to be contextualized with prior works like [A]. If my understanding is incorrect, then the hypothesis must be made clearer in the writeup.
3. **Tokenization Weaknesses**: The tokens as produced by EqualInfoAC are less semantic than SentencePiece. This also runs somewhat counter to traditional thinking of how attention operates in these sequences. Also, the tokenization produced by EqualInfoAC seems to be less stable. It would have been good to compare against tokenizations produced by standard AC and also stability of standard AC. There might be connections here to learnability, but it does not quite come out clearly from the text. Also, if possible, it would be contextualize the result with the prior works like [B]

[A] Are Transformers universal approximators of sequence-to-sequence functions?, ICLR 2020
[B] Getting the most out of your tokenizer for pre-training and domain adaptation, ICML 2024

---

### Author Response · Authors · 2024-11-09
**Author Response**

We thank all three reviewers for their insightful reviews. We appreciate their depth and attention to detail, especially considering the length of our paper. In line with reviewer suggestions, we’ve made a number of changes, which we believe have made the paper stronger. We’ve uploaded a new revision that includes the changes listed below.

**Reviewers**
* [R1] nNhF
* [R2] Kqn7
* [R3] BCNv

**Changes**
* Per R1 — Abstract and §4: Qualified our statements about sequence length and latency to highlight that the reduction depends on the resources available to optimize the per-step latency. The §4 changes are under “Equal-Info Windows make AC Learnable”.
* Per R3 — §2.5: Added discussion of Dagan et al. (2024) and Schmidt et al. (2024), near end of section.
* Per R2 — §2.5: Added discussion of Rajaraman et al. (2024) and Makkuva et al. (2024), as the last paragraph. The failure modes they observe on synthetic data are similar to the uniform distribution failure modes we observe.
* Per R1 — §3.3: Updated discussion of the AC “precision” hyper-parameter.
* Per R2 — §3.4: Formalized our tokenization process following Rajaraman et. al. (2024).
* Per R1 — §3.6: Clarified that the ByT5 tokenizer is performing a trivial UTF-8 encoding operation.
* Per R1 — §3.6: Specified that our subword baseline uses the Unigram algorithm as implemented by the SentencePiece library.
* Per R1 — §3.6: Added discussion of other subword tokenizers (BPE, WordPiece) and motivated the choice to use Unigram as our sole subword baseline. Previous work finds performance differences between Unigram / BPE / WordPiece are often statistically insignificant, and where there is signal, SentencePiece Unigram performs the best, at least in the English-only setting.
* Per R3 — §4: Added discussion of Yun et. al. (2020), as second paragraph.
* Per R1 — §4: Under “Equal-Info Windows make AC Learnable”, added for clarity: “on average SentencePiece tokenization requires 23% longer sequences to encode the same text”.
* Per R1 — §5.2: Added Figure 7 to help illustrate the seq2seq tasks used to test the model’s abilities to learn Arithmetic compression/decompression.
* Per R3 — §6.1: Added new analysis (Figure 8 and first two paragraphs) comparing the tokenization stability between EqualInfoAC and AC tokenized text. We test how modifying a prefix word affects subsequent tokenization, using edit distance, and show that AC tokenized text is even less stable than EqualInfoAC tokenized text.
* Per R3 — §6.1: Updated to clarify that when analyzing tokenization, we use “semantic” to refer to how well the tokens align to linguistic units (words and morphemes), as opposed to the higher-level semantics captured by attention later in the model.
* Per R1 — §6.2: Added motivation for our definition of “trival” accuracy, and clarified our choice to track training for EqualInfoAC[b=128] in Figure 8 (now 10) in footnote 22.

**Additional Discussion**
* R1 — We’re glad you find Figure 7 (now 9) illustrative, and we agree it would be nice to move it earlier in the paper. However, we found that so much needs to be explained first that moving it up caused confusion.
* R1 — With respect to “closing the gap”, Figure 3 is already a log-plot. Please let us know if you have any other suggestions on how to present closing the gap more clearly.

---

> ### Comment · Reviewer_nNhF · 2024-11-17
>
> Thanks for addressing all the comments and especially adding details in §5.2. I don't have any further suggestions and recommend accepting the paper.

---

### Decision · Action_Editor_Uw5K · 2024-12-03

**Recommendation:** Accept as is

**Comment:**

The paper studies the ability to train LLMs over compressed language data. Essentially switching the tokenizer for a compressed format. This is interesting because it allows one to essentially stretch the context window.   It also improves the inference speed and reduces computational costs, which will be beneficial on mobile phones or any device that is resource-constrained. But this also has come at the cost of a slightly lower  resulting perplexity. The reviewers, who I selected independently, were all unanimously in favour of the paper, and all independently recognized the strengths of the paper, and all strongly recommend the paper appear in ICLR.

**Audience:**

The audience would include those interested in using compression techniques for NLP, those who study tokenizers and are interested in novels ways for training LLMs.

**Claims And Evidence:**

The paper claims to study the ability to train LLMs on compressed data. They carefully expose how traditional compression techniques based on Arithmetic Coding can/often fail, and even how excessive compression makes learning impossible. The authors design a new compression technique that chunks together compressed text according to the number bytes, They show their new technique combined with an LLM makes learning possible, albeit at the cost of a slightly lower resulting perplexity.